# Brown adipose tissue and skeletal muscle coordinately contribute to thermogenesis in mice

Yuna Izumi-Mishima[1], Rie Tsutsumi[1], Tetsuya Shiuchi[2], Saori Fujimoto[1], Momoka Taniguchi[1], Mizuki Sugiuchi[1], Manaka Tsutsumi[1], Yuko Okamatsu-Ogura[3], Takeshi Yoneshiro[4], Masashi Kuroda[1], Kazuhiro Nomura[1], Hiroshi Sakaue[1]*

[1]Department of Nutrition and Metabolism, Institute of Biomedical Sciences, Tokushima University Graduate School, Tokushima, Japan; [2]Department of Integrative Physiology, Institute of Biomedical Sciences, Tokushima University Graduate School, Tokushima, Japan; [3]Laboratory of Biochemistry, Faculty of Veterinary Medicine, Hokkaido University, Hokkaido, Japan; [4]Division of Molecular Physiology and Metabolism, Tohoku University Graduate School of Medicine, Sendai, Japan

*For correspondence:
hsakaue@tokushima-u.ac.jp

Competing interest: The authors declare that no competing interests exist.

## eLife Assessment

This is a **useful** paper regarding the roles of brown adipose tissue and skeletal muscle in thermogenesis in mice, with potential significance for the field. The overall approach is innovative but on balance the evidence for the claim is **incomplete**, as cast immobilization, while innovative, is likely stressful, may impact muscle and BAT directly, and imposes an energetic cost of motion on the animal that is not accounted for. Further experiments are also needed to directly assess the role of adipose-derived BCAAs in thermogenesis. The authors have done a good job of textually editing their manuscript to clarify the findings and limitations of the study.

**Abstract** Endotherms increase the rate of metabolism in metabolic organs as one strategy to cope with a decline in the temperature of the external environment. However, an additional major contributor to maintenance of body temperature in a cold environment is contraction-based thermogenesis in skeletal muscle. Here, we show that impairment of hind limb muscle contraction by cast immobilization induced a loss of function of skeletal muscle and activated brown adipose tissue (BAT) thermogenesis as a compensatory mechanism. BAT utilizes free branched-chain amino acids (BCAAs) derived from skeletal muscle as an energy substrate for thermogenesis, and interleukin-6 released by skeletal muscle stimulates BCAAs production in muscle for support of BAT thermogenesis. Additionally, this thermoregulatory system between BAT and skeletal muscle may also play an important role in response to cold temperatures or acute stress. Our findings suggest that BAT and skeletal muscle cooperate to maintain body temperature in endotherms.

## Introduction

Endotherms rely on the regulation of heat generation by metabolism in tissues of the body to maintain a stable core body temperature in the face of a fluctuating temperature of the external environment and thereby to ensure optimal physiological function (*Tansey and Johnson, 2015*). Small mammals adjust their metabolic rate to cope with temperature fluctuations and to support their survival during seasonal acclimation. Some mammalian species are able to lower their metabolic rate and enter a state of torpor or hibernation in order to conserve energy and ensure survival (*Piscitiello et al., 2021*).

However, small mammals that do not hibernate must increase their rate of metabolism, including obligatory thermogenesis in metabolic organs such as the liver and skeletal muscle, as well as regulatory thermogenesis (shivering or nonshivering thermogenesis) in skeletal muscle and bBAT, in order to maintain thermal homeostasis in a cold environment (*Tansey and Johnson, 2015*; *Cannon and Nedergaard, 2004*; *Nowack et al., 2017*).

Under normothermic conditions, core body temperature is determined by basal metabolic organs such as the brain, heart, liver, and skeletal muscle (*Tansey and Johnson, 2015*). Skeletal muscle is the largest weight-bearing organ in humans and consumes a substantial amount of energy, even at rest. Shivering is the major source of heat production in acute cold exposure before a sufficient amount of UCP1 (uncoupling protein 1) can be recruited in BAT (*Tansey and Johnson, 2015*; *Rowland et al., 2015a*; *Golozoubova et al., 2001*). Shivering is gradually replaced by UCP1-dependent thermogenesis, and nonshivering thermogenesis is necessary for long-term adaptation to a cold environment (*Rowland et al., 2015a*). Skeletal muscle and BAT thermogenesis can compensate in part each other to maintain core body temperature is widely recognized (*Rowland et al., 2015a*; *Golozoubova et al., 2001*; *Janovska et al., 2023*; *Blondin et al., 2017*), and a recent study suggests that suppression of BAT thermogenesis has been found to promote thermogenesis in skeletal muscle (*Blondin et al., 2017*).

While skeletal muscle is the critical metabolic organ, it also stores an abundance of amino acids as energy substrates in the form of muscle proteins (*Bertile et al., 2021*). Skeletal muscle also has the plasticity for adapting to environmental conditions and shows metabolic flexibility in response to the substrate requirements of other organs, and thereby contributes to the maintenance of energy balance (*Bertile et al., 2021*; *Holeček, 2018*). Activation of BAT thermogenesis is associated with increased lipid and glucose catabolism (*Heine et al., 2018*; *Khedoe et al., 2015*; *Olsen et al., 2019*). Succinate and BCAAs have also been found to support BAT thermogenesis, however (*Mills et al., 2018*; *Yoneshiro et al., 2019*; *Yoneshiro et al., 2021*). Although succinate may be provided by muscle contraction, (*Mills et al., 2018*; *Reddy et al., 2020*) the primary source of BCAAs for BAT thermogenesis has remained unclear. Indeed, mitochondrial BCAAs catabolism in brown adipocytes promotes systemic BCAAs clearance. Skeletal muscle may, therefore, also respond to the energy demands of BAT.

Whereas body temperature and energy metabolism in metabolic organs are closely related, how they are regulated in a coordinated manner is poorly understood. Several endocrine factors alter metabolism in abdominal organs (*de Oliveira Dos Santos et al., 2021*). In particular, interleukin (IL)–6 is an endocrine factor that plays multiple roles in the response of the body to infection, exercise, and stress, as well as in inflammation (*Tanaka et al., 2014*; *Pedersen and Febbraio, 2008*; *Qing et al., 2020*). IL-6 induces various fever responses, including nonshivering thermogenesis in BAT, by promoting prostaglandin E2 (PGE2) synthesis in central vascular endothelial cells (*Chen et al., 2004*). On the other hand, IL-6 directly affects energy metabolism in various organs, including skeletal muscle, liver, and adipose tissue (*Serrano et al., 2008*; *Ellingsgaard et al., 2011*; *Knudsen et al., 2014*; *van Hall et al., 2003*; *Nonogaki et al., 1995*). These multiple roles suggest that IL-6 is a key molecule linking thermogenesis and energy metabolism.

Here, we show that a loss of skeletal muscle function by cast immobilization in mice activates a compensatory thermoregulatory system dependent on BAT thermogenesis. We also show that free BCAAs derived from skeletal muscle support BAT thermogenesis, and we identify IL-6 as a key regulator of this system. Our findings reveal the importance of skeletal muscle as a source of amino acids and uncover a previously unrecognized thermoregulatory system mediated by BAT and skeletal muscle metabolism.

## Results

### Cast immobilization activates a compensatory thermoregulatory system

We first examined the role of skeletal muscle in endothermic homeostasis by performing an acute cold tolerance test at 4 °C for 4 hr in mice with both hind limbs immobilized in casts. We confirmed that ~50% of systemic skeletal muscle was immobilized (*Supplementary file 1A*), and that the immobilization induced skeletal muscle atrophy within 3–5 days (*Figure 1—figure supplement*

*1A-1C*). Systemic locomotor activity was decreased during the dark phase in cast-immobilized mice compared to control mice (*Figure 1—figure supplement 1D and E*). Hence, the core body temperature showed a tendency to decrease in cast-immobilized mice compared to control mice under room temperature (*Figure 1—figure supplement 1F*). Mice with both hind limbs immobilized for 7 days showed a significantly lower core body temperature after cold exposure compared with control mice (*Figure 1A*). Cast immobilization for 24 hr also induced cold intolerance with preservation of muscle mass (*Figure 1B*, *Figure 1—figure supplement 1A-1C*). Expression of *Ucp1* mRNA in interscapular BAT (iBAT) was significantly increased after cold exposure in control mice, whereas those expression were not induced by cold exposure in cast immobilized mice (*Figure 1—figure supplement 1G*). Although the amount of *Sln* (sarcolipin) mRNA in immobilized muscle was not increased by acute cold exposure, expression of *Ucp2*, *Ucp3,* and *Ppargc1a* (PGC-1α) were significantly increased in control mice but not in cast-immobilized mice (*Figure 1*, *Figure 1—figure supplement 1H-1K*). To investigate whether cold intolerance depends on the immobilized muscle mass, we examined the change in core body temperature in mice with unilateral immobilization. Systemic locomotor activity during cold exposure was similar for mice with unilateral or bilateral immobilization and for control mice (*Figure 1*, *Figure 1—figure supplement 1L*). However, the decrease in body temperature induced by cold exposure in mice with unilateral immobilization was attenuated compared with that in mice subjected to bilateral immobilization (*Figure 1C*). These findings suggested that immobilization of skeletal muscle suppresses thermogenesis in this organ independently of systemic locomotor activity.

To investigate the mechanism underlying the maintenance of core body temperature in mice with cast immobilization, we examined the effects of such immobilization on skeletal muscle and BAT. Metabolomic profiling of the immobilized muscle revealed decreases in the amounts of fumaric acid and malic acid within 1 day of cast immobilization (*Figure 1—figure supplement 2A*). The amount of *Ucp3* mRNA increased transiently in immobilized muscle before returning to baseline levels, whereas that of *Ucp2* mRNA was not affected until 5 days of cast immobilization (*Figure 1—figure supplement 2B and 2C*). Sarcolipin signaling has been shown to promote mitochondrial biogenesis and oxidative metabolism via activation of $Ca^{2+}$- and calmodulin-dependent protein kinase II (CaMKII) and PGC-1α in skeletal muscle (*Maurya et al., 2018*). However, we found that expression of the *Sln*, *Camk2a*, *Ppargc1a* (PGC-1α), and *Tfam* (transcription factor A, mitochondrial) genes was not increased in immobilized muscle (*Figure 1—figure supplement 2D-2G*). These findings thus suggested that nonshivering thermogenesis was induced minimally in cast-immobilized skeletal muscle. On the other hand, expression of UCP1 at the mRNA and protein levels as well as the abundance of PGC-1α, a transcriptional cofactor that promotes the expression of thermogenesis-related genes, showed transient increases in iBAT after cast immobilization (*Figure 1E–G*). Small and multilocular lipid droplets were apparent in adipocytes of iBAT within 24 hr of cast immobilization (*Figure 1—figure supplement 2H*). However, mice subjected to cast immobilization showed no increase in food consumption or systemic activity levels (*Figure 1—figure supplements 1D and 1E and 2I*). These results indicated that the suppression of muscle thermogenesis by cast immobilization induces BAT thermogenesis without an increase in food consumption or systemic activity levels. Cast immobilization thus preferentially activated nonshivering thermogenesis in BAT, not in skeletal muscle.

## Cast immobilization promotes BAT thermogenesis via sympathetic activation

We next explored the mechanism underlying the activation of BAT thermogenesis by cast immobilization. We first focused on the sympathetic nervous system, a general inducer of BAT thermogenesis, (*Bartness et al., 2010*) and found that the norepinephrine concentration in iBAT was tended to increase transiently after 24 hr of cast immobilization (*Figure 1H*). We then subjected mice to sympathetic denervation of iBAT. Expression of *Ucp1* mRNA in iBAT was significantly attenuated by such denervation in mice with or without subsequent cast immobilization (*Figure 1I*). Whereas adipocytes of iBAT showed small and multilocular lipid droplets after 7 days of cast immobilization in sham-operated mice, iBAT of mice subjected to surgical denervation was hypertrophied and did not show such lipid droplets (*Figure 1—figure supplement 2J*). Moreover, mice with denervated iBAT manifested a lower core body temperature compared with sham-operated mice after cold exposure at 4 °C for 1 hr, and cast immobilization further increased the extent of hypothermia in the denervated

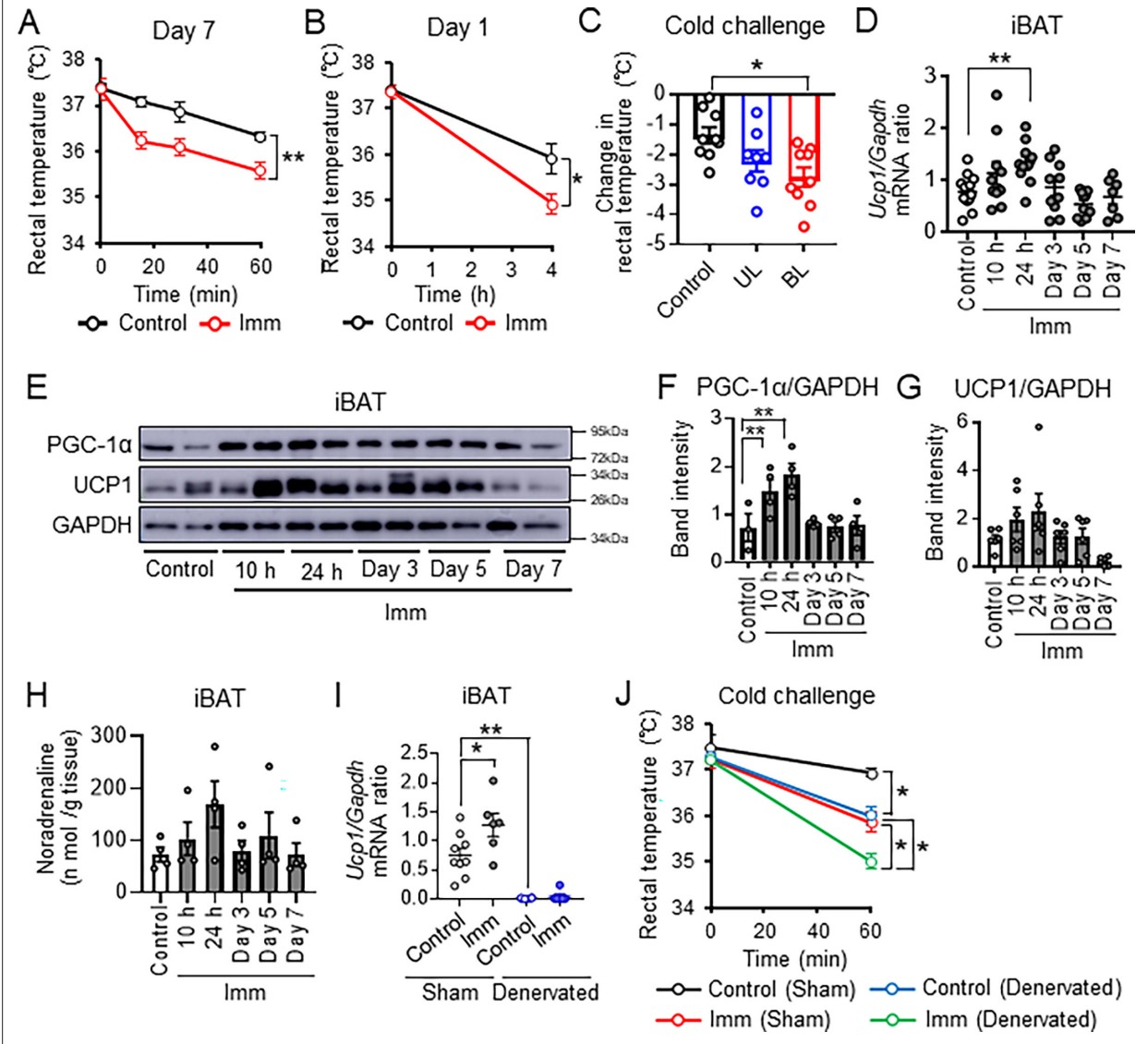

**Figure 1.** Cast immobilization induces brown adipose tissue (BAT) thermogenesis via sympathetic activation. (**A**) Time course of rectal core body temperature of cast-immobilized (gypsum, n=6) and control (n=5) mice subjected to a cold (4 °C) challenge test. The test was performed 7 days after bilateral immobilization of hind limbs. (**B**) Rectal core body temperature of cast-immobilized (n=5) and control (n=5) mice during cold (4 °C) exposure at 24 hr after bilateral cast immobilization. (**C**) Decline in rectal temperature of cast-immobilized and control (n=9) mice after cold (4 °C) exposure for 4 hr performed 2 hr after unilateral (UL, n=8) or bilateral (BL, n=9) immobilization. (**D**) Reverse transcription (RT) and real-time polymerase chain reaction (PCR) analysis of the *Ucp1/Gapdh* mRNA abundance ratio in iBAT of control mice and mice subjected to bilateral cast immobilization for the indicated times (n=7–13 per group). (**E-G**) Immunoblot analysis of PGC-1α (n=3 or 4 per group) and UCP1 (n=6 per group) in interscapular BAT (iBAT) of control mice and mice subjected to bilateral cast immobilization for the indicated times. GAPDH was examined as a loading control. Representative blots (**E**) and densitometric quantitation of the PGC-1α/GAPDH (**F**) and UCP1/GAPDH (**G**) band intensity ratios are shown. (**H**) Noradrenaline concentrations in iBAT of cast-immobilized (n=4) and control (n=4) mice after subsequent bilateral cast immobilization. (**I**) RT and real-time PCR analysis of the *Ucp1/Gapdh* mRNA abundance ratio in iBAT subjected to surgical denervation or sham surgery, either in control mice or in mice at 24 hr after subsequent bilateral cast immobilization (n=6–9 per group). (**J**) Rectal core body temperature of mice as in (**I**) subjected to cold (4 °C) exposure for 1 hr (n=3 or 4 per group). All quantitative data are means ± SEM. *p<0.05, **p<0.01 as determined by two-way ANOVA followed by Tukey's post hoc test or the post hoc paired/unpaired t-test with Bonferroni's correction (**A, B, I, and J**), by one-way ANOVA followed by Tukey's post hoc test (**C**), by Dunnett's test (**D, F, G, and H**), or by the unpaired t-test (**I**). See also (*Figure 1—figure supplements 1 and 2*, and *Supplementary file 1A*).

The online version of this article includes the following source data and figure supplement(s) for figure 1:

**Source data 1.** Cast immobilization induces brown adipose tissue (BAT) thermogenesis via sympathetic activation.

**Source data 2.** *Figure 1E—F* Raw WB data.

**Source data 3.** Labeled WB data.

*Figure 1 continued on next page*

*Figure 1 continued*

**Figure supplement 1.** Cold intolerance in cast-immobilized mice independent of muscle mass and locomotor activity.

**Figure supplement 2.** Activation of BAT thermogenesis in cast immobilized at room temperature.

mice (*Figure 1J*). These results suggested that BAT thermogenesis is activated via sympathetic nerves to maintain core body temperature in cast-immobilized mice.

## Cast immobilization alters systemic metabolic dynamics associated with BAT thermogenesis

We next examined the metabolic dynamics of iBAT as well as systemic metabolic changes in mice subjected to cast immobilization. Previous study suggests that acute cold exposure in mice increases the free amino acids concentration and carbohydrate metabolites such as the glycolysis pathway, the pentose phosphate pathway, and the tricarboxylic acid (TCA) cycle in BAT (*Okamatsu-Ogura et al., 2020*). Consistent with previous findings, metabolomics analysis revealed increases in the amounts of carbohydrate metabolites and amino acids in iBAT that were apparent as early as 10 hr after cast immobilization, and several glucose and amino acid metabolic pathways were significantly activated (*Figure 2—figure supplement 1A*). Increases in these metabolites after cast immobilization were not apparent in mice with iBAT denervation (*Figure 2A and B*). Brown adipocytes incorporate fatty acids and various other energy substrates to initiate thermogenesis and contribute to the maintenance of core body temperature (*Wang et al., 2021*). We found that expression of the genes for fatty acid (*Figure 2C*) and succinate (*Figure 2D*) transporters in iBAT was induced by cast immobilization and that such induction was prevented by iBAT denervation. The concentration of succinate was also increased in iBAT of cast-immobilized mice in a manner sensitive to iBAT denervation (*Figure 2E*). Consistent with these changes in metabolite levels, oxygen consumption and systemic lipid utilization were increased by cast immobilization in intact mice (*Figure 2F and G*) but not in those subjected to iBAT denervation (*Figure 2—figure supplement 1B and C*).

The weight of epididymal white adipose tissue (eWAT) and hepatic glycogen content were reduced by cast immobilization (*Figure 2H and I*). Expression of the genes for glucose-6-phosphatase (G6Pase) and phosphoenolpyruvate carboxykinase (PEPCK) in the liver increased gradually after cast immobilization, peaking at day 5 (*Figure 2—figure supplement 2A and B*), suggesting that gluconeogenesis was slowly activated in the cast-immobilized mice. Metabolomic profiling of the liver revealed marked increases in the amounts of 3-phosphoglyceric acid and 2-phosphoglyceric acid within 1 day of cast immobilization (*Figure 2—figure supplement 2C*), suggesting that glycerol influx was also enhanced in the liver. Together, these results thus indicated that cast immobilization affects metabolic dynamics in the liver and WAT, and that these changes might occur after the activation of BAT thermogenesis.

The serum concentration of noradrenaline was increased within 1 day of cast immobilization (*Figure 2—figure supplement 2D*), but these metabolic changes induced by cast immobilization were not associated with by an increase in the serum concentration of corticosterone or in expression of the gene for corticotropin-releasing hormone (CRH) in the hypothalamus (*Figure 2—figure supplement 1E and F*). These results suggest that cast immobilization may activate BAT thermogenesis and systemic metabolic changes through lower body temperature resulting from suppression of muscle thermogenesis rather than stress.

## Free amino acids are transferred from skeletal muscle to BAT and the liver for energy homeostasis

We next focused on the metabolism of amino acids in BAT and skeletal muscle in mice after cast immobilization. As shown above, amino acid concentrations in iBAT were increased by cast immobilization in a manner sensitive to denervation of iBAT (*Figure 2B*, *Figure 2—figure supplement 1A*). The amounts of free amino acids were also increased in both soleus and extensor digitorum longus (EDL) muscles of cast-immobilized mice (*Figure 3A*), whereas those in serum and the liver were not increased by cast immobilization (*Figure 3B*). Consistent with previous findings (*Kawanishi et al., 2018*), we found that gene expression for the ubiquitin ligases Atrogin-1 and MuRF-1 was increased during the early phase (days 1–3) of cast immobilization (*Figure 3—figure supplement 1A and B*). We then examined gene expression profiles for solute carrier (SLC) transporters that mediate the

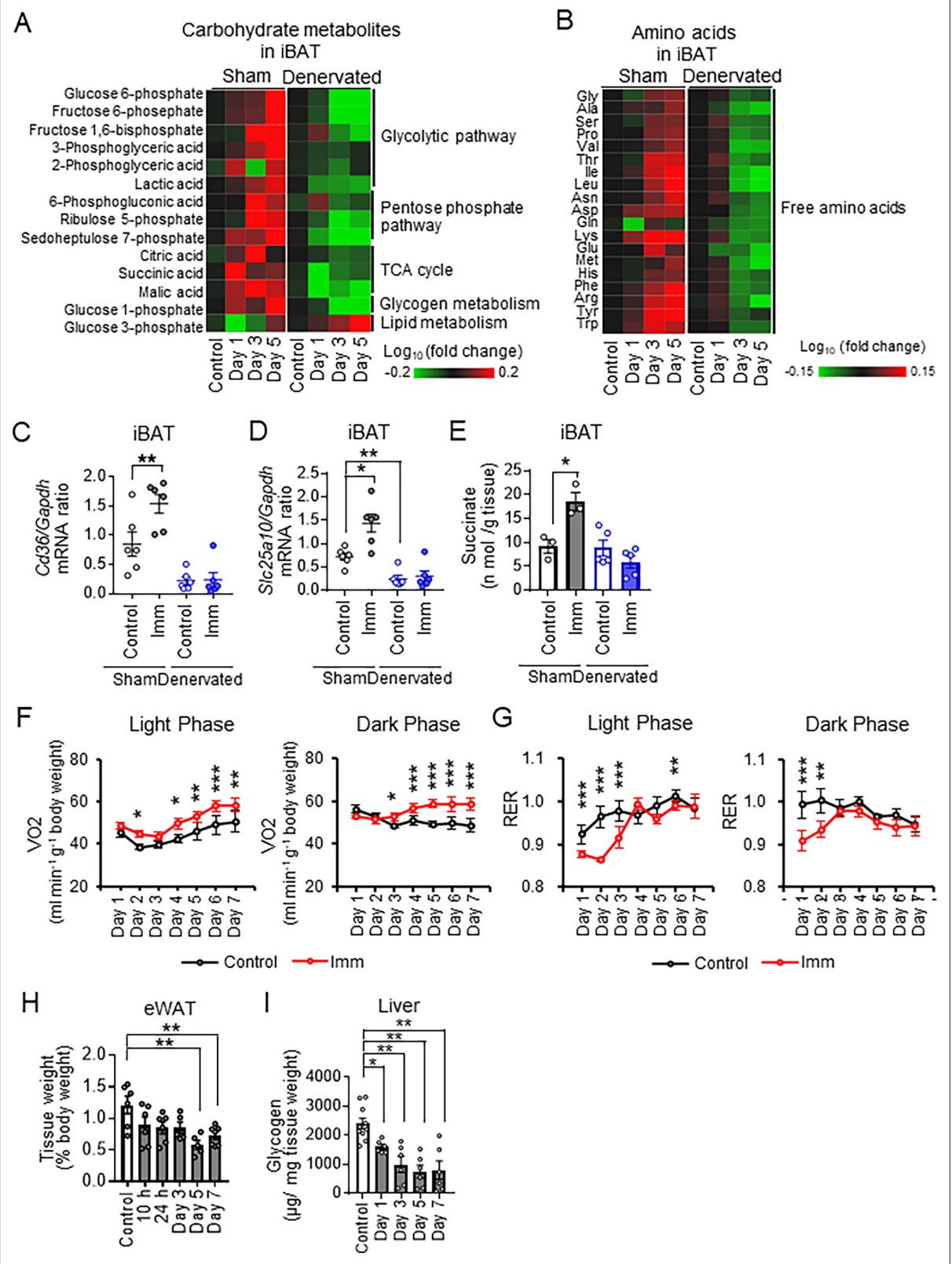

**Figure 2.** Cast immobilization alters systemic metabolic dynamics associated with brown adipose tissue (BAT) thermogenesis. (**A and B**) Concentrations of carbohydrate metabolites (**A**) and amino acids (**B**) in sham-operated or denervated interscapular BAT (iBAT) of mice without (control) or at the indicated times after bilateral cast immobilization. The metabolomics data are presented as heat maps corresponding to the log$_{10}$ value of fold change in cast-immobilized mice relative to the corresponding control mice and are means of four or five mice in each group. TCA, tricarboxylic acid cycle. (**C**

*Figure 2 continued on next page*

*Figure 2 continued*

**and D**) Reverse transcription (RT) and real-time polymerase chain reaction (PCR) analysis of mRNA abundance for the fatty acid transporter CD36 and the succinate transporter SLC25A10 in sham-operated or denervated iBAT of control mice or mice subjected to bilateral cast immobilization for 24 hr (n=6 per group). (**E**) Succinate content of sham-operated or denervated iBAT of control mice or mice subjected to bilateral cast immobilization for 24 hr (n=3–5 per group). (**F and G**) Oxygen consumption rate (VO$_2$) and respiratory exchange ratio (RER), respectively, for control mice and mice subjected to bilateral cast immobilization for 7 days (n=6 per group). (**H**) Weight of eWAT for control mice and mice subjected to bilateral cast immobilization for the indicated times (n=5–7 per group). (**I**) Hepatic glycogen content for control mice and mice subjected to bilateral cast immobilization for the indicated times (n=6–10 per group). Date in (**C**) through (**I**) are means ± SEM. *p<0.05, **p<0.01 as determined by one-way ANOVA followed by Tukey's post hoc test (**C-E**), by two-way ANOVA followed by Tukey's post hoc test or the unpaired post hoc t-test with Bonferroni's correction (**F and G**), or by Dunnett's test (**H and I**). See also (*Figure 2—figure supplements 1 and 2*).

The online version of this article includes the following source data and figure supplement(s) for figure 2:

**Source data 1.** Cast immobilization alters systemic metabolic dynamics associated with brown adipose tissue (BAT) thermogenesis.

**Figure supplement 1.** Metabolic alterations in BAT after cast immobilization.

**Figure supplement 2.** Metabolic changes in the liver following BAT thermogenesis in cast-immobilized mice.

transport of amino acids across cell membranes, as well as for mitochondrial BCAA catabolic enzymes (BCAT2, branched-chain aminotransferase 2; BCKDHA, branched-chain keto acid dehydrogenase E1 subunit α) in iBAT, liver, and soleus. Expression of the genes for SLC1A5 (a sodium-dependent glutamate transporter) and SLC38A2 (a sodium-coupled neutral amino acid transporter) was significantly increased in iBAT after cast immobilization for 10 hr (*Figure 3—figure supplement 1C-H*). In the liver, expression of the gene for SLC38A2, which is upregulated by extracellular amino acid deprivation, (*Gazzola et al., 2001*; *Gjymishka et al., 2008*) and of that for BCKDHA, which is thought to be upregulated under fasting conditions to oxidize BCAAs for gluconeogenesis, (*Huang and Chuang, 1999*; *Kobayashi et al., 1999*) was significantly increased by cast immobilization (*Figure 3—figure supplement 1I-M*). On the other hand, gene expression for amino acid transporters (SLC1A5, SLC7A5, SLC38A2) and BCAAs catabolic enzymes (BCAT2, BCKDHA) in soleus was significantly decreased after cast immobilization (*Figure 3—figure supplement 1N-S*). Expression of the gene for SLC43A1, which is upregulated by food deprivation and is thought to transport amino acids from cells to the external environment (*Fukuhara et al., 2007*), was significantly increased in soleus by cast immobilization (*Figure 3—figure supplement 1Q*).

We next investigated amino acid incorporation in tissues by administration of [3H] leucine into the tail vein of mice. Incorporation of [3H] leucine was not increased in skeletal muscle of mice subjected to cast immobilization for 1 or 3 days (data not shown), whereas it was significantly increased in iBAT, liver, and kidney of those subjected to cast immobilization for 1 day (*Figure 3C*). Gene expression for SLC25A44, a mitochondrial BCAAs transporter that contributes to BAT thermogenesis (*Yoneshiro et al., 2019*), was increased in iBAT at 24 hr after cast immobilization (*Figure 3D*, *Figure 3—figure supplement 2A-C*). However, the early increases in expression of the genes for SLC25A44, SLC1A5, and SLC38A2 induced by cast immobilization in iBAT of intact mice were prevented by denervation of iBAT (*Figure 3D*, *Figure 3—figure supplement 1D-J*). Our results thus suggested that free amino acids are utilized as an energy substrate in BAT and the liver rather than in skeletal muscle of cast-immobilized mice.

## Cast immobilization upregulates IL-6 gene expression in BAT and skeletal muscle

We next set out to identify regulatory molecules that might contribute to the maintenance of body temperature and to the systemic metabolic changes in cast-immobilized mice. Gene expression analysis revealed that the IL-6 gene tended to be induced earlier and to a greater extent in iBAT and soleus than were genes for other cytokines including tumor necrosis factor–α (TNF-α), monocyte chemoattractant protein–1 (MCP-1), IL-1β, IL-10, and IL-15; for batokines including fibroblast growth factor 21 (FGF21) and bone morphogenetic protein 8B (BMP8B); or for myokines including FGF21 and irisin (*Figure 4A-D*, *Figure 4—figure supplement 1A-N*). Whereas the increase in the amount of *Il6* mRNA was apparent as early as 6–10 hr in iBAT, that in soleus was more gradual (*Figure 4C and D*). IL-6 has been found to be released by various organs, including adipose tissue and skeletal muscle, in response to inflammation, stress, or exercise (*de Oliveira Dos Santos et al., 2021*; *Tanaka et al., 2014*; *Pedersen and Febbraio, 2008*; *Qing et al., 2020*). The abundance of *Il6* mRNA in liver, eWAT,

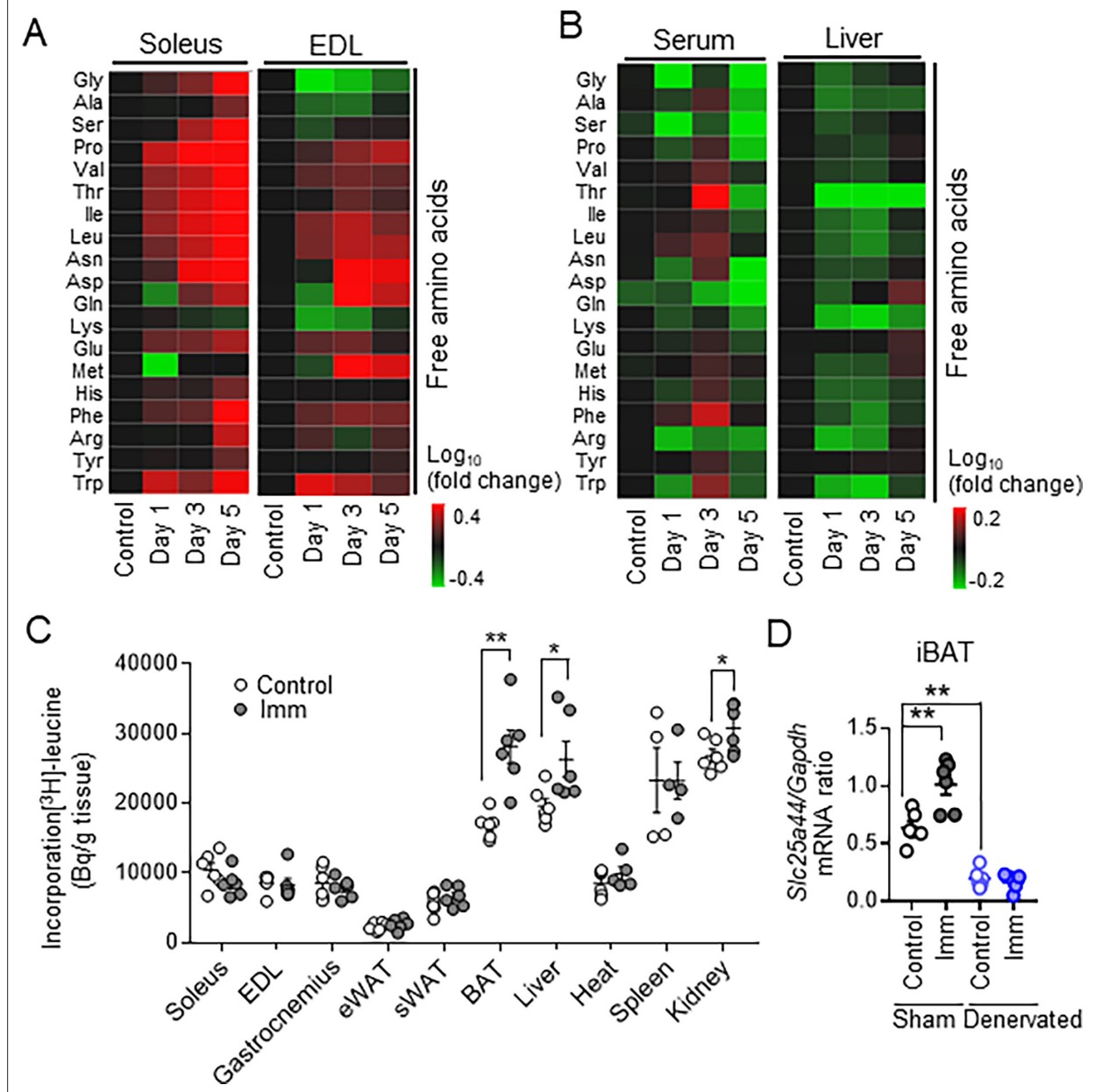

**Figure 3.** Free amino acids are transferred from skeletal muscle to brown adipose tissue (BAT) and the liver for maintenance of energy homeostasis. (**A and B**) Metabolomics analysis of amino acid concentrations in soleus and extensor digitorum longus (EDL) muscles (**A**) as well as in serum and the liver (**B**) of control mice and mice subjected to bilateral cast immobilization for the indicated times. Results are presented as heat maps of the $\log_{10}$ value of fold change for cast-immobilized mice relative to control mice and are means of three mice per group. (**C**) Organ-specific [³H] leucine uptake in control mice and mice subjected to bilateral cast immobilization for 24 hr (n=4–6 per group). sWAT, subcutaneous white adipose tissue (WAT). (**D**) Reverse transcription (RT) and real-time polymerase chain reaction (PCR) analysis of *Slc25a44* expression in sham-operated or denervated interscapular BAT (iBAT) of control mice and mice subjected to subsequent bilateral cast immobilization for 24 hr (n=6 per group). Data in (**D**) and (**E**) are means ± SEM. *p<0.05, **p<0.01 as determined by the unpaired t-test (**C**) or by one-way ANOVA followed by Tukey's post hoc test (**D**). See also (***Figure 3—figure supplements 1 and 2***).

The online version of this article includes the following source data and figure supplement(s) for figure 3:

**Source data 1.** Free amino acids are transferred from skeletal muscle to brown adipose tissue (BAT) and the liver for maintenance of energy homeostasis.

**Figure supplement 1.** Transcriptional reprogramming in BAT, liver, and soleus muscle following cast immobilization.

**Figure supplement 2.** Gene expression change in denervated BAT after cast immobilization.

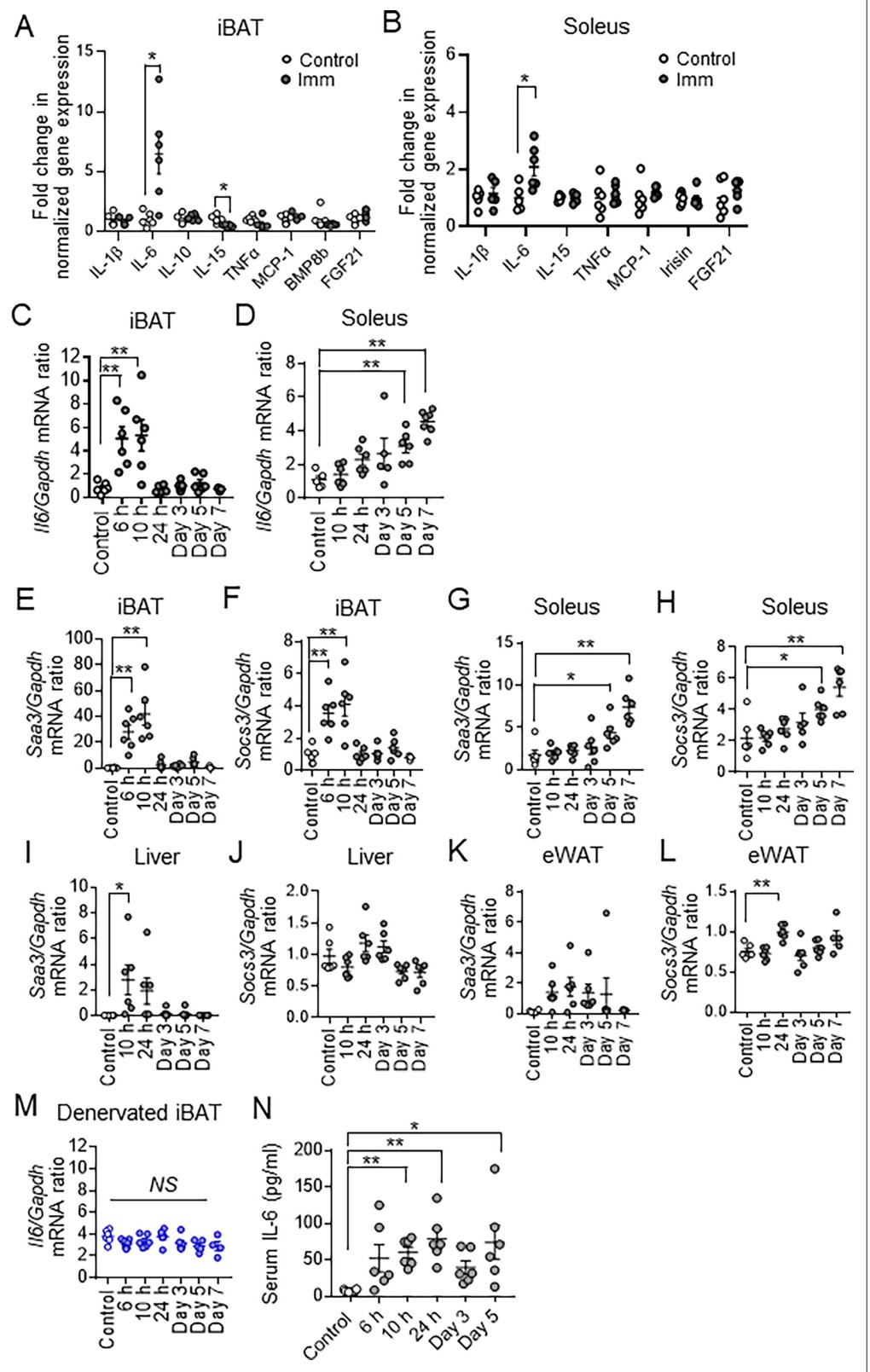

**Figure 4.** Cast immobilization upregulates *Il6* expression in brown adipose tissue (BAT) and skeletal muscle. (**A and B**) Reverse transcription (RT) and real-time polymerase chain reaction (PCR) analysis of *Il1b*, *Il6*, *Il10*, *Il15*, *Tnfa*, *Mcp-1*, *Bmp8b*, *Fgf21*, and *Irisin* expression in interscapular BAT (iBAT) (**A**) or soleus (**B**) of control mice or mice at 10 hr (iBAT) or 24 hr (soleus) after bilateral cast immobilization (n=5 or 6 per group).(**C and D**) RT and real-time PCR

*Figure 4 continued on next page*

*Figure 4 continued*

analysis of *Il6* expression in iBAT (**C**) and soleus (**D**) of control mice and mice at the indicated times after bilateral cast immobilization (n=3–7 per group). (**E-L**) RT and real-time PCR analysis of *Saa3* (E, G, I, and K) and *Socs3* (F, H, J, and L) mRNA abundance in iBAT (**E and F**), soleus (**G and H**), liver (**I and J**), and epididymal white adipose tissue (eWAT) (**K and L**) of control mice and mice at the indicated times after bilateral cast immobilization (n=3–7 per group) (**M**) RT and real-time PCR analysis of *Il6* expression, respectively, in denervated iBAT of control mice or mice at the indicated times after bilateral cast immobilization (n=4–7 per group). (**N**) Serum IL-6 concentration in control mice or mice at the indicated times after bilateral cast immobilization (n=6 per group). All data are means ± SEM. *p<0.05, **p<0.01, NS (not significant) as determined by the unpaired t-test (**A and B**) or by Dunnett's test (**C–N**). See also (***Figure 4—figure supplement 1***).

The online version of this article includes the following source data and figure supplement(s) for figure 4:

**Source data 1.** Cast immobilization upregulates *Il6* expression in brown adipose tissue (BAT) and skeletal muscle.

**Figure supplement 1.** Gene expression levels of secretory factors in BAT and soleus muscle following cast immobilization.

---

or spleen was not affected by cast immobilization (***Figure 4—figure supplement 1O-Q***). Expression of the genes for serum amyloid A3 (*Saa3*) and suppressor of cytokine signaling 3 (*Socs3*), both of which are regulated by IL-6 signaling, was also increased in iBAT and soleus at the same time as was that of *Il6* (***Figure 4E–H***). Expression of these genes was also increased in the liver or eWAT at 10 or 24 hr after cast immobilization (***Figure 4I–L***). However, expression of *Il6* was not induced by cast immobilization in denervated iBAT (***Figure 4M***). The serum concentration of IL-6 was significantly increased from 10 hr after cast immobilization (***Figure 4N***), whereas that of TNF-α was not altered in cast-immobilized mice (***Figure 4—figure supplement 1R***). Together, these results suggested that cast immobilization induces *Il6* expression in iBAT and skeletal muscle, and that induction of *Il6* expression by cast immobilization in iBAT is dependent on the sympathetic nervous system.

## IL-6 affects energy metabolism in BAT and skeletal muscle of cast-immobilized mice

We investigated the role of IL-6 in the skeletal muscle–BAT thermoregulatory system with the use of subjected IL-6 knockout (KO) mice to cast immobilization. Expression of *Ucp1* in iBAT was significantly increased at 24 hr after cast immobilization in both WT and IL-6 KO mice (***Figure 5—figure supplement 1A***). However, the core body temperature of cast-immobilized IL-6 KO mice was decreased to a greater extent by cold exposure at 4 °C for 4 hr compared with that of cast-immobilized WT mice (***Figure 5A***). Furthermore, unlike that in WT mice, oxygen consumption was not increased in IL-6 KO mice after cast immobilization (***Figure 5—figure supplement 1B and C***). Metabolomics analysis revealed that the amounts of carbohydrate metabolites, including glucose 1-phosphate, glucose 6-phosphate, and phosphoenolpyruvic acid, in iBAT were higher for IL-6 KO mice than for WT mice and were not increased substantially by cast immobilization in the mutant mice (***Figure 5B***). Whereas the concentrations of amino acids such as Ser, Asn, Lys, Met, His, Phe, and Tyr in iBAT of WT mice were increased markedly at 24 hr after cast immobilization, such was not the case for IL-6 KO mice (***Figure 5C***). Expression of the genes for SLC25A44, BCKDHA, and CD36 in iBAT was higher for IL-6 KO mice than for WT mice, but expression of the genes for SLC25A44 and CD36 in iBAT of IL-6 KO mice was not increased further at 24 hr after cast immobilization (***Figure 5D–F***). Incorporation of [³H] leucine was also not increased in iBAT of IL-6 KO mice after cast immobilization for 24 hr (***Figure 5G***). In addition, adipocytes with small and multilocular lipid droplets were observed in iBAT of IL-6 KO mice with or without cast immobilization (***Figure 5—figure supplement 1D***). The respiratory exchange ratio (RER) of WT mice was significantly decreased in both the light and dark phases for the first 5 days of cast immobilization, whereas similar differences were apparent only for the first 2 days in IL-6 KO mice (***Figure 5—figure supplement 1E and F***). IL-6 has been shown to promote lipolysis and fatty acid oxidation in WAT (***van Hall et al., 2003***). We found that, unlike in WT mice, cast immobilization did not result in a loss of eWAT mass in IL-6 KO mice (***Figure 5—figure supplement 1G***). Collectively, these data suggested that utilization of energy substrates was not substantially altered by cast immobilization in IL-6 KO mice.

The amounts of BCAAs in soleus were significantly increased after cast immobilization in WT mice but not in IL-6 KO mice (***Figure 5H-J***). Serum BCAA concentrations were decreased after cast

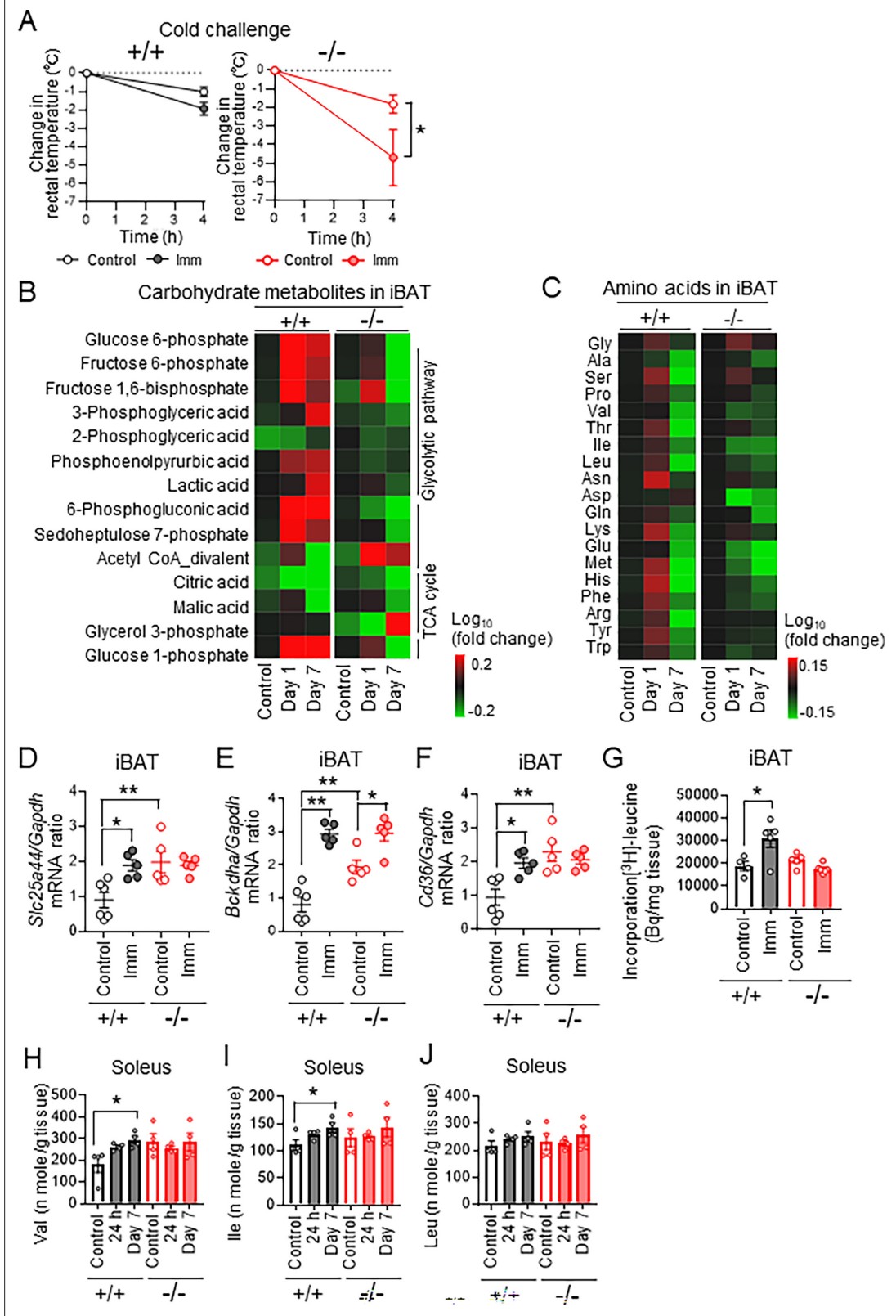

**Figure 5.** Interleukin (IL)-6 affects energy metabolism in brown adipose tissue (BAT) and skeletal muscle of cast-immobilized mice. (**A**) Change in rectal body temperature during cold exposure at 4 °C for 4 hr for wild-type (WT) (n=4 or 5) and interleukin (IL)-6 knockout (KO) (n=4 or 5) mice that had been subjected (or not) to bilateral cast immobilization for 24 hr. (**B and C**) Concentrations of carbohydrate metabolites (**B**) and amino acids (**C**) in interscapular BAT (iBAT) of IL-6 KO (–/–) and WT (+/+) mice without (control) or with bilateral cast immobilization for the indicated times. Results are

*Figure 5 continued on next page*

*Figure 5 continued*

presented as heat maps of the $\log_{10}$ value of the fold change relative to control WT mice and are means of four mice in each group. (**D–F**) Reverse transcription (RT) and real-time polymerase chain reaction (PCR) analysis of the expression of *Slc25a44* (**D**), *Bckdga* (**E**), and *Cd36* (**F**) in iBAT of IL-6 KO and WT mice without (control) or with bilateral cast immobilization for 24 hr (n=5 or 6). (**G**) Incorporation of [³H] leucine in iBAT of IL-6 KO and WT mice without (control) or with bilateral cast immobilization for 24 hr (n=4 or 5 per group). (**H–J**) Branched-chain amino acid (BCAA) (Val, Ile, and Leu, respectively) concentrations in soleus of IL-6 KO and WT mice without (control) or with bilateral cast immobilization for 1 or 7 days (n=4 per group). Data in (**A**) and (**D**) through (**J**) are means ± SEM. *$p<0.05$, **$p<0.01$ as determined by one-way ANOVA followed by Tukey's post hoc test (**D–G**), by Dunnett's test (**H–J**), or by two-way ANOVA followed by Tukey's post hoc test (**A**). See also (*Figure 5—figure supplement 1*).

The online version of this article includes the following source data and figure supplement(s) for figure 5:

**Source data 1.** Interleukin (IL)-6 affects energy metabolism in brown adipose tissue (BAT) and skeletal muscle of cast-immobilized mice.

**Figure supplement 1.** Effect of IL-6 on metabolic changes in cast-immobilized mice.

immobilization in both WT and IL-6 KO mice (*Figure 5—figure supplement 1H-J*). These findings suggested that deficiency of IL-6 impairs the maintenance of core body temperature and alters energy metabolism in BAT and skeletal muscle in mice subjected to cast immobilization.

## Administration of IL-6 increases BCAA concentrations in skeletal muscle of cast-immobilized IL-6 KO mice

We examined whether administration of exogenous IL-6 might increase amino acid concentrations in skeletal muscle and BAT of IL-6 KO mice with cast immobilization. Administration of IL-6 restored the increase in the amounts of amino acids, including BCAAs in soleus of IL-6 KO mice with cast immobilization (*Figure 6A-C*, *Figure 6—figure supplement 1A*). Treatment with IL-6 also increased expression of *Saa3* and *Socs3* (*Figure 6—figure supplement 1B and C*) as well as the amounts of BCAAs (*Figure 6D*) in cultured mouse C2C12 myotubes. However, the amounts of carbohydrate metabolites in iBAT of cast-immobilized IL-6 KO mice were not increased by IL-6 administration (*Figure 6—figure supplement 1D*). In contrast to its effects on soleus, administration of IL-6 also did not increase the amounts of amino acids in iBAT of IL-6 KO mice with cast immobilization (*Figure 6—figure supplement 1A and E*). In addition, exogenous IL-6 did not increase the expression of UCP1, SLC25A44, and BCKDHA genes in iBAT of these mice (*Figure 6—figure supplement 1F-H*). On the other hand, administration of IL-6 ameliorated the cold intolerance of IL-6 KO mice with cast immobilization (*Figure 6E*). We, therefore, examined whether IL-6 might increase core body temperature via the sympathetic nervous system with the use of C57BL/6 J mice with denervated iBAT. Core body temperature of cold-exposed mice with cast immobilization and denervated iBAT was not significantly increased by administration of IL-6 (*Figure 6F*).

We also found that the core body temperature of IL-6 KO mice was lower than that of WT mice after cold exposure at 4 °C for 4 hr (*Figure 6—figure supplement 1I*) whereas the amount of *Ucp1* mRNA in iBAT was significantly increased for IL-6 KO mice compared with WT mice at room temperature (*Figure 6—figure supplement 1J*). We then investigated the possible direct effects of IL-6 on energy metabolism in BAT with the use of cultured mouse brown adipocytes. Expression of the SLC25A44, BCKDHA, and UCP1 genes in the brown adipocytes were not increased by treatment with IL-6 (*Figure 6—figure supplement 1K-M*). In contrast, treatment with the β₃-adrenergic receptor agonist CL316 243 increased the expression of these genes, and this upregulation was suppressed by treatment with IL-6 (*Figure 6—figure supplement 1K-M*). Long-term administration of CL316 243 in mice was previously shown to increase expression of *Ucp1*, *Slc25a44*, and *Bckdha* in BAT (*Yoneshiro et al., 2021*). We found that administration of CL316 243 for 7 days significantly increased the amounts of *Ucp1*, *Slc25a44*, and *Bckdha* mRNAs in iBAT of IL-6 KO mice but not in that of WT mice (*Figure 6—figure supplement 1N-P*). Together, these data suggested that IL-6 may suppress β-adrenergic receptor–mediated thermogenesis and BCAA metabolism in brown adipocytes. Nevertheless, IL-6 induces fever response through stimulation of the central (*Qing et al., 2020*; *Chen et al., 2004*) and uptake of energy-rich BCAA delivered to BAT from skeletal muscle may facilitate a net induction of BAT thermogenesis.

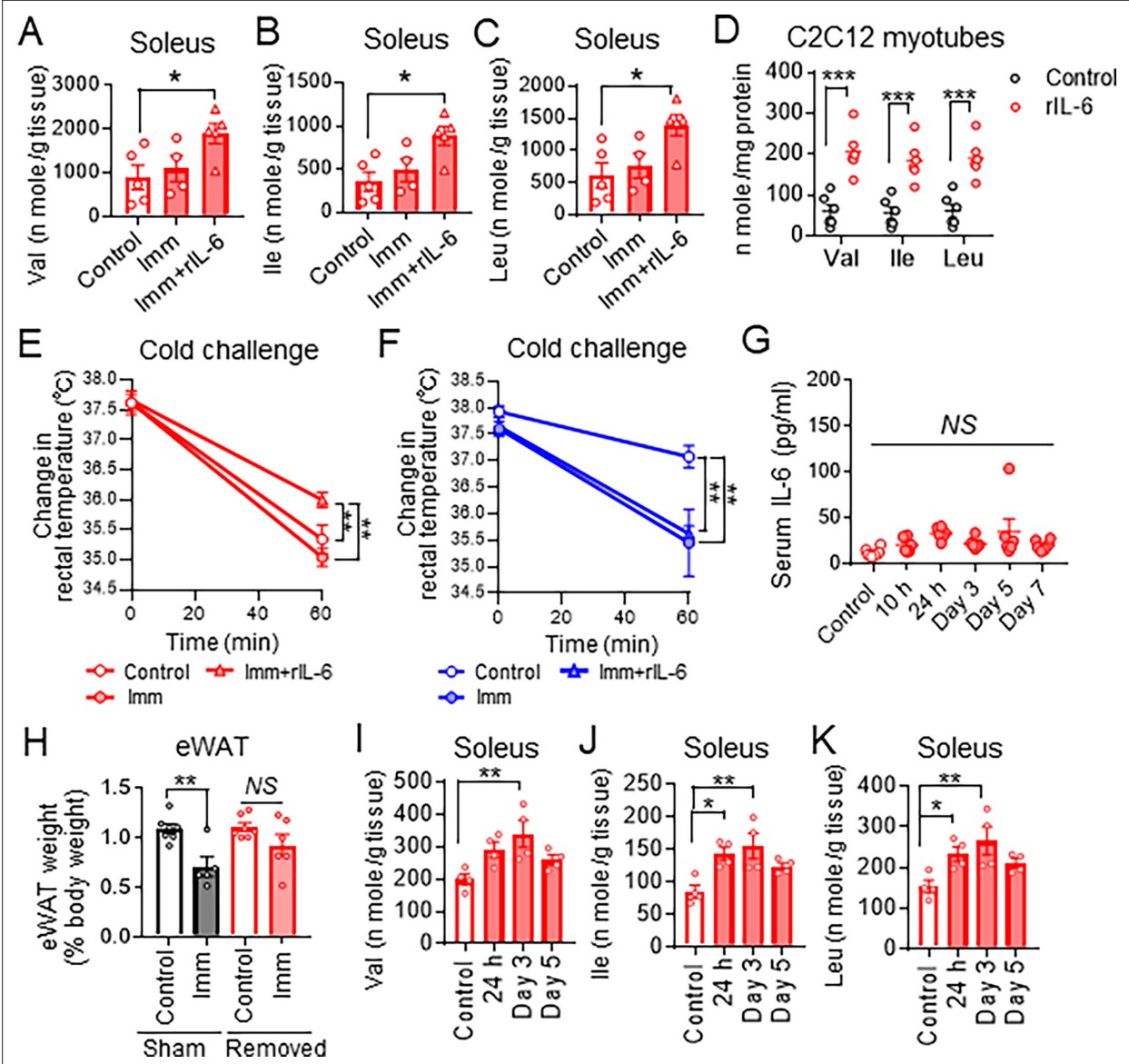

**Figure 6.** Skeletal-muscle-derived interleukin (IL)-6 increases branched-chain amino acid (BCAA) concentrations in skeletal muscle for support of brown adipose tissue (BAT) thermogenesis. (**A–C**) Concentrations of BCAAs (Val, Ile, and Leu, respectively) in the soleus of control IL-6 knockout (KO) mice and at 3 hr after intraperitoneal administration of recombinant IL-6 (rIL-6, 400 ng) or vehicle in IL-6 KO mice that had been subjected to bilateral cast immobilization for 24 hr (n=4 or 5 per group). (**D**) BCAA concentrations in C2C12 myotubes incubated in the absence or presence of rIL-6 (50 ng/ml) for 1 hr (n=6 independent experiments). (**E**) Rectal core body temperature during cold exposure at 4 °C for 1 hr for control IL-6 KO mice and for IL-6 KO mice subjected to bilateral cast immobilization for 24 hr and injected intraperitoneally with rIL-6 (400 ng) or vehicle 30 min before cold challenge (n=5 per group). The white circles represent control mice, the red circles indicate cast-immobilized mice, and the red triangles represent cast-immobilized mice that were treated with recombinant IL-6 (rIL-6). (**F**) Rectal core body temperature during cold exposure at 4 °C for 1 hr for C57BL/6 J mice with denervated iBAT subjected or not to bilateral cast immobilization and treatment with rIL-6 or vehicle as in (**E**) (n=4 per group). (**G**) Serum IL-6 concentration of mice subjected to surgical removal of interscapular BAT (iBAT) and then subjected (or not, control) to bilateral cast immobilization for the indicated times (n=4–6 per group). The white circles represent control mice, the blue circles indicate C57BL/6 J mice with denervated iBAT, and the blue triangles denote C57BL/6 J mice with denervated iBAT that were treated with recombinant IL-6 (rIL-6). (**H**) Weight of epididymal white adipose tissue (eWAT) in mice subjected to surgical removal of iBAT (or to sham surgery) followed (or not, control) by bilateral cast immobilization for 7 days (n=5–7 per group). (**I-K**) Concentrations of BCAAs (Val, Ile, and Leu, respectively) in soleus of mice subjected to surgical removal of iBAT and then subjected (or not, control) to bilateral cast immobilization for the indicated times (n=4 per group). All data are means ± SEM. *p<0.05, **p<0.01, NS (not significant) as determined by Dunnett's test (A–C and G and I-K), or by the unpaired t-test (**D**) or by one-way ANOVA followed by Tukey's post hoc test (**H**), or by two-way ANOVA followed by the post hoc paired/unpaired t-test with Bonferroni's correction (**E and F**). See also (***Figure 6—figure supplements 1 and 2***, and ***Supplementary file 1B***).

*Figure 6 continued on next page*

*Figure 6 continued*

The online version of this article includes the following source data and figure supplement(s) for figure 6:

**Source data 1.** Skeletal-muscle-derived interleukin (IL)-6 increases branched-chain amino acid (BCAA) concentrations in skeletal muscle for support of brown adipose tissue (BAT) thermogenesis.

**Figure supplement 1.** Effect of IL-6 on amino acid metabolism in myocytes and brown adipocytes.

**Figure supplement 2.** Changes in skeletal muscle gene expression in cast-immobilized mice after surgical removal of BAT.

## Skeletal muscle-derived IL-6 increases BCAA concentrations in skeletal muscle for BAT thermogenesis

Given that our data suggested that an elevated blood concentration of IL-6 contributes to maintenance of body temperature in cast-immobilized mice in a manner dependent on the sympathetic nervous system (*Figure 6E and F*), we sought to determine the main source of circulating IL-6 in cast-immobilized mice. We first focused on the acute expression of the IL-6 gene apparent in iBAT after cast immobilization (*Figure 4C*), and we subjected mice to surgical removal of iBAT. The serum IL-6 concentration of such mice was not increased after cast immobilization (*Figure 6G*). We also found that the reduction in eWAT mass after cast immobilization in C57BL/6 J mice was suppressed by surgical removal of iBAT (*Figure 6H*). These results suggested that BAT-derived IL-6 induces lipolysis in WAT.

We further investigated the effects of BAT-derived IL-6 on skeletal muscle. Whereas the serum concentration of IL-6 in mice depleted of iBAT was not affected by cast immobilization (*Figure 6G*), the concentrations of BCAAs in soleus were increased after cast immobilization in such mice (*Figure 6I–K*). The amount of *Slc43a1* mRNA in soleus of mice depleted of iBAT was significantly increased by cast immobilization (*Figure 6—figure supplement 2A*). Gene expression for other amino acid transporters (such as SLC1A5 and SLC7A5) and for BCAA catabolic enzymes (BCAT2 and BCKDHA) in soleus also showed a similar pattern of changes after cast immobilization in iBAT-depleted mice (*Figure 6—figure supplement 2B-E*) as in intact mice (*Figure 3—figure supplement 1L, 1M, 1P and Q*). Moreover, cast immobilization for 7 days induced similar losses of both soleus and gastrocnemius mass in mice with or without removal of iBAT (*Figure 6—figure supplement 2F* and *Supplementary file 1B*). These data thus suggested that IL-6 secretion in an autocrine or paracrine manner may regulate BCAA metabolism in skeletal muscle independently of IL-6 derived from iBAT.

## Acute cold exposure or restraint stress also induced IL-6 from skeletal muscle to supply BCAA for BAT thermogenesis

We further found that expression of *Il6* was also increased in not immobilized fore limb muscles (biceps brachii muscle; BBM) after cast immobilization (*Figure 7—figure supplement 1A*). Expression of *Saa3* and *Socs3* were increased in BBM at the same time as was that of *Il6* after cast immobilization (*Figure 7—figure supplement 1B and 1C*). Furthermore, gene expression for the ubiquitin ligases Atrogin-1 and MuRF-1 in BBM was increased during the early phase of cast immobilization (*Figure 7—figure supplement 1D and E*). BCAA concentrations in BBM were also increased after cast immobilization (*Figure 7—figure supplement 1F and H*). These data suggest that cast immobilization stimulates BCAA metabolites in not immobilized fore limb muscles as well as immobilized muscle.

Finally, we investigated whether the thermoregulatory system through amino acid metabolism in the interaction between BAT and skeletal muscle also plays an important role in other conditions, such as cold temperature and restraint stress. Acute cold exposure increased expression of *Il6* only in iBAT and skeletal muscle in mice (*Figure 7A*), and expression of *Saa3* and *Socs3* were increased in soleus (*Figure 7—figure supplement 1I and J*). Gene expression for the ubiquitin ligases Atrogin-1 and MuRF-1 was not significantly increased in soleus (*Figure 7—figure supplement 1K and L*), but both soleus and gastrocnemius mass were decreased by cold exposure at 4 °C for 4 hr (*Figure 7—figure supplement 1M and N*). Furthermore, the amounts of valine in soleus were significantly increased by acute cold exposure in WT mice but not in IL-6 KO mice (*Figure 7B-D*, *Figure 7—figure supplement 1O*). BCAA concentrations in soleus were also increased after cold exposure at 4 °C for 4 hr in iBAT-depleted mice (*Figure 7—figure supplement 1P*).

Restraint stress for 6 hr on mice increased expression of *Ucp1* in iBAT (*Figure 7—figure supplement 2A*), and the amounts of *Il6* mRNA was increased in iBAT and skeletal muscle (*Figure 7E and*

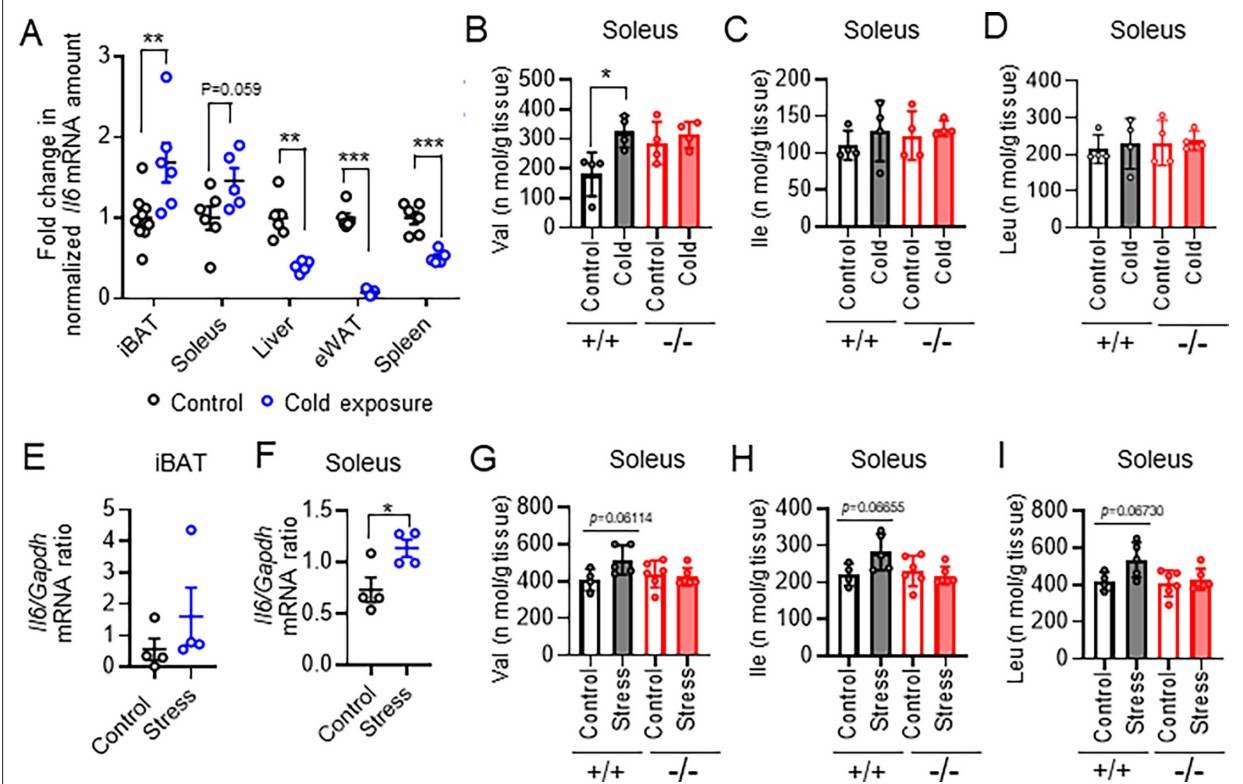

**Figure 7.** Acute cold exposure or restraint stress also induced interleukin (IL)-6 from skeletal muscle to supply branched-chain amino acid (BCAA) for brown adipose tissue (BAT) thermogenesis. (**A**) Reverse transcription (RT) and real-time polymerase chain reaction (PCR) analysis of *Il6* expression in interscapular BAT (iBAT), soleus, liver, epididymal white adipose tissue (eWAT), and spleen of mice subjected (or not) to cold (4 °C) exposure for 4 hr (control, n=6–9; cold exposure, n=5 or 6). (**B-D**) Concentrations of BCAAs (Val, Ile, and Leu, respectively) in soleus of wild-type (WT) and IL-6 knockout (KO) mice subjected (or not) to cold exposure at 4 °C for 4 hr (n=4 per group). (**E and F**) RT and real-time PCR analysis of *Il6* expression in iBAT (**E**) and soleus (**F**) of control mice and mice subjected to restraint stress for 6 hr (n=4 per group). (**G–I**) Concentrations of BCAAs (Val, Ile, and Leu, respectively) in soleus of WT and IL-6 KO mice subjected (or not) to restraint stress for 6 hr (n=4 per group). All data are means ± SEM. *p<0.05, **p<0.01, NS (not significant) as determined by the unpaired t-test (**A–I**). See also *Figure 7—figure supplements 1 and 2*.

The online version of this article includes the following source data and figure supplement(s) for figure 7:

**Source data 1.** Acute cold exposure or restraint stress also induced interleukin (IL)-6 from skeletal muscle to supply branched-chain amino acid (BCAA) for brown adipose tissue (BAT) thermogenesis.

**Figure supplement 1.** Acute cold exposure stimulates skeletal muscle IL-6 expression and BCAA efflux.

**Figure supplement 2.** Restraint stress stimulates skeletal muscle IL-6 expression and BCAA efflux.

*F*). Expression of *Saa3* and *Socs3* were tended to increase in soleus (*Figure 7—figure supplement 2B and C*). Skeletal muscle mass was not decreased by restraint stress for 6 hr (*Figure 7—figure supplement 2D-F*), but gene expression for the ubiquitin ligases Atrogin-1 and MuRF-1 were increased after restraint stress (*Figure 7—figure supplement 2G-H*). BCAA concentrations in soleus were tended to increased by restraint stress in WT mice but not in IL-6 KO mice (*Figure 7G-I*, *Figure 7—figure supplement 2I*). Together, thermoregulatory system through amino acid metabolism in skeletal muscle and BAT may be activated under cold temperature and acute stress.

## Discussion

We have shown here that cast immobilisation leads to a passive loss of skeletal muscle function, resulting in failure to maintain adequate core body temperature in a cold environment. Cast immobilization also activated BAT thermogenesis via the sympathetic nervous system and triggered systemic changes in energy metabolism associated with BAT thermogenesis. Furthermore, we found that free BCAAs derived from skeletal muscle serve as substrates for energy metabolism in BAT, and that

skeletal muscle–derived IL-6 promotes this provision of BAT with amino acids from muscle. In addition, this thermoregulatory system between BAT and skeletal muscle may also be activated in response to cold temperature or acute stress.

A first aim of our study was to investigate the role of skeletal muscle in thermogenesis. In this study, mice with impairment of hind limb muscle contraction by cast immobilization were used as a model for a loss of muscle function. Exercise-induced muscle contraction generates large amounts of heat which dependents on hydrolysis of ATP, and skeletal muscle, as the first organ to be recruited for thermogenesis, plays an important role in maintenance of body temperature in endotherms (*Rowland et al., 2015b*). Muscle thermogenesis can account for up to 90% of systemic oxygen consumption during periods of maximal recruitment of muscle contraction, such as during exercise or an intense bout of shivering (*Rowland et al., 2015b*; *Zurlo et al., 1990*). We found that cast immobilization decreased the amounts of metabolites in TCA cycle and suppressed expression of mitochondrial-related genes from the early stage of immobilization, suggesting that metabolic rate is rapidly decreased in immobilized muscle. On the other hands, thermoregulatory system in endotherms cannot be explained by thermogenesis based on muscle contraction alone, with nonshivering thermogenesis being required as a component of the ability to tolerate cold temperatures in the long term (*Tansey and Johnson, 2015*). Our results now show that expression of the gene for sarcolipin, a key regulator of the sarco/endoplasmic reticulum $Ca2^{+}$ ATPase (SERCA) and an important mediator of muscle nonshivering thermogenesis, was not increased in muscle after cast immobilization. In contrast, we have now shown that UCP3 gene expression in muscle was transiently increased during the early stage of cast immobilization. Although the importance of UCP3 for mitochondrial energy metabolism is well established, (*Bouillaud et al., 2016*; *Lombardi et al., 2019*), its physiological function as a mitochondrial uncoupler remains unclear (*Vidal-Puig et al., 2000*; *Shabalina et al., 2010*). In addition, we found that several genes induced by cold stimulation in skeletal muscle were not increased in cast-immobilized mice. Our results thus show that metabolic thermogenesis in skeletal muscle is crucial for mammals, and that cast immobilization increases cold intolerance, even in BAT-enriched mammals such as rodents. Although our study showed that cold intolerance in mice was observed after just 2 hr of cast immobilization, these results could also be attributed not only to loss of skeletal muscle function but also to stress, decreased calorie reserves, or reduced systemic locomotor activity.

In contrast to skeletal muscle, BAT thermogenesis was activated via the sympathetic nervous system under room temperature even when skeletal muscle was immobilized. It is a well-recognized fact that BAT and skeletal muscle function in an orchestrated manner to maintain core body temperature in endotherms (*Rowland et al., 2015a*; *Golozoubova et al., 2001*; *Janovska et al., 2023*). A bit surprisingly, the capacity of the BAT itself was insufficient to maintain stable body temperature during acute cold exposure. Our results also suggest that the increase in noradrenaline concentrations in BAT or in blood are transient, suggesting that the activation of sympathetic nerve activity after cast immobilization is also transient. Consequently, the expression of thermogenic genes and metabolic changes in BAT may also have been induced to peak after 24 hr of cast immobilization. In addition, it has been reported that long-term cast immobilization increases nonshivering thermogenesis in immobilized muscle (*Tomiya et al., 2019*). Our results show that the expression of *Ucp2* and *Sln* were increased in immobilized muscle after 7 days of cast immobilization. These data suggest that the maintenance of core body temperature by BAT thermogenesis may be an important system during short-term cast immobilization.

We also found that changes in systemic energy metabolism induced by cast immobilization were associated with the activation of BAT thermogenesis. Activation of BAT results in the tissue becoming a high consumer of a variety of energy substrates, including lipids, glucose, BCAAs, succinate, and lactate (*Wang et al., 2021*). Previous findings reported that mitochondrial BCAA catabolism in brown adipocytes promotes systemic BCAA clearance, suggesting that BCAAs may be supplied to BAT from other organs during BAT thermogenesis. Our results suggest that skeletal muscle is a source of free amino acids for BAT thermogenesis or hepatic gluconeogenesis. In response to metabolic demands imposed by starvation, exercise, or cold exposure, for example, skeletal muscle manifests metabolic flexibility in order to meet the demands of other organs and is an important determinant of metabolic homeostasis (*Bertile et al., 2021*; *Holeček, 2018*). In contrast, the reduction in metabolic rate may contribute to protein conservation and maintain skeletal muscle mass under hibernation (*Bertile et al., 2021*). Our findings now provide important insight into the role of skeletal muscle as a source

of amino acids. However, a recent study also suggests that a highly active proteolysis system in the heart provides substantial amounts of amino acids for distribution to other organs via the bloodstream (*Murashige et al., 2020*). The heart may, therefore, be another source of free BCAAs for BAT thermogenesis in cast-immobilized mice.

We found that muscle-derived IL-6 directly increased the abundance of BCAAs in skeletal muscle after cast immobilization. Previous studies have shown that excessive IL-6 induces amino acid catabolism in skeletal muscle, (*Bonetto et al., 2012*; *Zanders et al., 2022*), and that IL-6 is one of the factors responsible for the induction of muscle atrophy in cast-immobilized mice (*Hirata et al., 2022*). IL-6 is a secreted factor induced by several mechanisms, including muscle contraction, intracellular calcium signaling, and inflammation. (*Pedersen and Febbraio, 2008*; *Bustamante et al., 2014*) However, inflammation associated with macrophage infiltration is not thought to be induced in the early stages of muscle atrophy (*Kawanishi et al., 2018*), and thus inflammation-induced IL-6 expression may, therefore, not contribute to the changes in amino acid metabolism in cast-immobilized muscle. In addition, whereas intracellular calcium concentrations may increase after muscle immobilization for 2 weeks (*Tomiya et al., 2019*), they appear to decrease in response to short-term cast immobilization (*Hirata et al., 2022*). Cast immobilization-induced *Il6* expression may, therefore, also not be regulated by calcium signaling. Upregulation of IL-6 expression by cast immobilization was previously shown to be mediated by a Piezo1-KLF15 axis (*Hirata et al., 2022*), whereas our observation that *Il6* expression were increased in non-immobilized forelimb muscles. Furthermore, acute cold exposure and short-term restraint stress tended to increase *Il6* expression in skeletal muscle, suggesting that muscle-derived IL-6 for thermoregulatory system may also be regulated by the central nervous system. A recent study showed that motor circuits modulate the production of neutrophil-induced chemokines in skeletal muscle after acute stress (*Poller et al., 2022*). The regulation of IL-6 production in muscle by the central nervous system warrants further investigation.

The central nervous system tightly controls core body temperature through integrates information about external temperature, humidity, and thermal sensation to induce an adaptive response (*Tansey and Johnson, 2015*). Disappointingly, our study was inconclusive as to whether the trigger for BAT thermogenesis after cast immobilization was hypothermia associated with loss of skeletal muscle function or stress. However, we found that acute cold exposure and short-term restraint stress may also recruits substrate supply from skeletal muscle for BAT thermogenesis. Possibly, thermoregulatory system through amino acid metabolism in skeletal muscle and BAT may also be an important metabolic strategy even in the case of fever response induced by stress or infection. Furthermore, inflammation, infection, and acute stress trigger rapid increases in the circulating IL-6 concentration (*Qing et al., 2020*; *Cheng et al., 2015*) and induce a fever response by promoting PGE2 synthesis. Mitochondrial BCAA oxidation in BAT was recently shown to be increased to support thermogenesis induced by PGE2 (*Yoneshiro et al., 2019*). We found that BAT-derived IL-6 increased blood IL-6 levels after cast immobilization and administration of exogenous IL-6 ameliorated cold intolerance in cast-immobilized IL-6 KO mice via the sympathetic nervous system. It may modulate thermal and energy homeostasis through the fever response and metabolic regulation in multiple organs.

In conclusion, we have shown that cast immobilization-induced thermogenesis in BAT that was dependent on the utilization of free amino acids derived from skeletal muscle, and that muscle-derived IL-6 stimulated BCAA metabolism in skeletal muscle. Although some effects, such as activation of BAT thermogenesis and changes in energy metabolic dynamics, observed with cast immobilization were modest in magnitude, thermoregulatory system through amino acid metabolism in skeletal muscle and BAT may be an important metabolic strategy even under cold temperature or acute stress. Our findings may provide new insights into the significance of skeletal muscle as a large reservoir of amino acids in the regulation of body temperature. In addition, given that circulating levels of IL-6 or BCAAs are associated with obesity and diabetes,(*Wallenius et al., 2002*; *Wang et al., 2011*) IL-6–mediated BCAA metabolism in skeletal muscle and BAT may also be associated with muscle atrophy in metabolic diseases. Further investigation of IL-6–dependent amino acid metabolism in BAT and skeletal muscle may inform the development of preventive or therapeutic interventions for some forms of muscle atrophy.

## Limitations of the study

In our cast immobilization strategy, we were unable to determine which components of muscle thermogenesis are actually inhibited by cast immobilization, and what is the relative heat contribution. This study does not experiment that directly tests whether BCAAs derived from adipose tissue are used for thermogenesis, which would require more robust tracing experiments.

In addition, given that rodents are BAT-enriched mammals, whether BAT thermogenesis is similarly activated in humans after skeletal muscle immobilization remains to be investigated. Our study also did not determine the mechanism underlying the induction of *Il6* expression in skeletal muscle by cast immobilization. In addition, whereas muscle-derived IL-6 may stimulate protein catabolism in immobilized muscle, the mechanism by which IL-6 increases BCAA concentrations in skeletal muscle remains unclear. In further investigation, RNA-seq profiling of BAT and muscle tissues should be used to rigorously determine the mechanisms of BCAA metabolism stimulated by muscle-derived IL-6.

# Materials and methods

## Mice

All experiments were approved by the Animal Care and Use Committee of Tokushima University (protocol numbers: T2023-17 and T2020-63) and were conducted in accordance with the guidelines for the care and use of animals approved by the Council of the Physiological Society of Japan. Every effort was made to minimize animal suffering and to reduce the number of animals used in the experiments. All mice were housed at a constant room temperature of 23°±1 °C and with a 12-hr-light/12-hr-dark cycle (lights on at 8.00 a.m.). They were fed standard non-purified chow (Oriental Yeast) and had free access to both food and water. Our study examined male mice because male animals exhibited less variability in phenotype. Male mice at 9–13 weeks of age were studied. The mice were randomly assigned to experimental groups at the time of purchase or weaning. C57BL/6 J mice were obtained from Japan SLC (Shizuoka, Japan), and IL-6 KO mice (B6;129S2-Il6<tm1kopf>) from RIKEN BRC through the National Bioresource Project of MEXT/AMED (*Wallenius et al., 2002*). The IL-6 KO mice were maintained on the C57BL/6 J background. All animal experiments were conducted in accordance with the ARRIVE guidelines.

## Cells

The mouse myoblast cell line C2C12 was obtained from American Type Culture Collection (CRL-1772) and was maintained in Dulbecco's modified Eagle's medium (DMEM) containing 25 mM glucose (#D6429, Sigma) and supplemented with 10% fetal bovine serum (535–94155, BioSera) and 1% penicillin-streptomycin (15140–122, Gibco). After the cells had achieved 95 to 100% confluence, the culture medium was changed to DMEM containing 25 mM glucose and supplemented with 2% horse serum (16050–130, Thermo Fisher Scientific) and the cells were cultured for 5 days to promote their differentiation from myoblasts into myotubes. The myotubes were washed with phosphate-buffered saline (PBS) and then incubated in DMEM containing 25 mM glucose and supplemented with 1% bovine serum albumin (A8806-5G, Sigma) and 1% penicillin-streptomycin for 16 hr before exposure to recombinant mouse IL-6 (rIL-6) at 50 ng/ml (575702, BioLegend) or vehicle (PBS) in the same medium for 1 hr. The treated cells were washed with PBS or 5% mannitol solution and collected for RNA or metabolite extraction.

Mouse preadipocytes were obtained from Dr. Takeshi Yoneshiro at the University of Tohoku, Tohoku, Japan (*Shinoda et al., 2015*) and were maintained in DMEM containing 25 mM glucose and supplemented with 10% fetal bovine serum and 1% penicillin-streptomycin. After the cells had achieved 95 to 100% confluence, they were induced to differentiate into brown adipocytes by incubation in the same medium that was also supplemented with 1 nM triiodothyronine (45006–44, Nacalai Tesque), insulin (093–06473, Wako) at 5 µg/ml, dexamethasone (11107–51, Nacalai Tesque) at 2 µg/ml, 0.5 mM isobutylmethylxanthine (15879, Sigma), and 125 µM indomethacin (19233–51, Nacalai Tesque). After culturing for 2 days, the medium was replaced with DMEM containing 25 mM glucose and supplemented with 10% fetal bovine serum, 1% penicillin-streptomycin, and insulin (5 µg/ml), and the cells were maintained in this medium with medium replenishment every 2 days. The cells were fully differentiated at 6–7 days after the induction of differentiation. For experiments, the brown adipocytes were washed with PBS at 7 days after differentiation initiation and then incubated in DMEM

containing 25 mM glucose and supplemented with 1% bovine serum albumin and 1% penicillin-streptomycin for 12 hr before exposure to rIL-6 (50 ng/ml), 1 μM CL316 243 (Ab144605, Abcam), both agents, or vehicle (PBS). The treated cells were collected after 24 h for RNA extraction.

Cell lines were originally authenticated by the manufacturer, ATCC, through STR profiling. ATCC also verified that the cells were free from mycoplasma contamination. All cell lines used in this study were obtained from reliable sources, confirmed by morphology and proliferation characteristics, and tested negative for mycoplasma. None are listed among the commonly misidentified cell lines published by the International Cell Authentication Committee (ICAC).

## Cast immobilization
Mice were subjected to bilateral or unilateral cast immobilization of hind limbs, and unfixed mice were studied as controls. In brief, mice were lightly anesthetized with inhalational isoflurane (MSD Animal Health), and the hind limbs were fixed in a natural position, with care taken to avoid skin irritation and congestion. After casting, the mice were housed individually with free access to standard mouse chow and water. The mice out of cast immobilization were excluded from experiments. In all experiments, mice were killed by cardiac puncture after food deprivation for 3 hr and tissue samples were collected. Proportion of immobilized skeletal muscle weight for cast-immobilized mice was obtained by extirpation skeletal muscle of the posterior cervical region and measuring wet tissue weights.

## Surgical denervation of iBAT
Surgical denervation of iBAT was performed on 9-week-old male mice as described previously (*Vaughan et al., 2014*). Animals were anesthetized by intraperitoneal injection of domitor (Zenoaq) at a dose of 0.75 mg/kg, midazolam (Wako) at 4 mg/kg, and butorphanol tartrate (Meiji Seika Pharma) at 5 mg/kg, and the incision site was shaved and then disinfected with 0.05% chlorhexidine gluconate (5% Hibitane, Sumitomo Pharma). A midline skin incision was made in the interscapular region to expose both iBAT pads, the nerve fibers (five nerves per pad) were identified and cut with sterile scissors, and the incision was closed with sutures. For sham surgery, a skin incision was made and the medial side was gently exposed. All mice were then individually housed in clean cages at room temperature and were monitored daily. Cast immobilization was performed 1 week after surgery.

## Surgical removal of iBAT
Surgical removal of iBAT was performed on 11-week-old male mice, with cast immobilization being performed 1 week after surgery. Animals were anesthetized and the incision site was shaved and disinfected as described for iBAT denervation. A midline skin incision was made in the interscapular region, all iBAT was removed, and the incision was closed with sutures. For sham surgery, a skin incision was made and the medial side was gently exposed. All mice were then housed individually in clean cages at room temperature.

## Acute cold tolerance test
Core body temperature was measured with a Homeothermic Monitor (BWT-100A, BRC) by gentle insertion of a thermal probe into the rectum of the mouse. After recording the baseline body temperature at room temperature, animals were placed in a cold chamber at 4 °C for up to 4 hr without access to food. Tissue was then collected and snap-frozen.

## Acute restraint stress
Mice were confined in 50 mL tubes with several ventilation holes of about 0.5 cm in diameter on the sides, and were restrained for 6 hr without access to food. Mice restrained in tubes were maintained in a state in which they could hardly move their bodies. In all experiments, mice were killed by cardiac puncture and tissue samples were collected.

## Treatment of mice with CL316 243 or rIL-6
IL-6 KO mice with cast immobilization for 24 hr were subjected to intraperitoneal administration of 400 ng of rIL-6 (575702, BioLegend) in saline or of saline alone. The mice were maintained sedentary for 3 hr without access to food before collection of tissue samples. Acute exposure to a cold environment chamber was initiated 30 min after intraperitoneal treatment with rIL-6 (400 ng) or saline in

IL-6 KO mice or in C57BL/6 J mice with denervated iBAT that had been subjected to cast immobilization for 24 hr. IL-6 KO mice with denervated iBAT were treated intraperitoneally with CL316 243 (Ab144605, Abcam) at a dose of 0.1 mg/kg, rIL-6 (400 ng), both agents, or saline vehicle, and tissue was collected 3 hr later. For long-term treatment of IL-6 KO mice with CL316 243, the animals were injected intraperitoneally with the drug (0.1 mg/kg per day) or saline for 7 days. Tissue was collected at 3 hr after drug treatment on the last day.

### Oxygen consumption and RER measurement

Mice were acclimated to the analysis cage for 3 days before the start of metabolic measurements. They were housed individually with free access to food and water. Analysis of respiratory gases was performed with a mass spectrometer (ARCO-2000, ARCO System). Measurements were performed for seven consecutive days after cast immobilization.

### Locomotor activity recording

The activity level of mice was assessed with an infrared activity monitor (ACTIMO-100N, Shin Factory). A cage containing one mouse was placed inside the activity monitor with infrared beams at 20 mm intervals. Mouse movements were counted every 0.5 s for 24 hr. Recording was performed for seven consecutive days after cast immobilization.

### RNA isolation and RT and real-time PCR analysis

Total RNA was extracted from mouse tissues and cells with the use of RNAiso (#9109, Takara Bio) and subjected to RT with the use of a TaKaRa PrimeScript II first Strand cDNA Synthesis Kit (Takara Bio). The resulting cDNA was subjected to real-time PCR analysis with Fast SYBR Green Master Mix (3485612, Applied Biosystems) in a Step One Plus Real-Time PCR System (Applied Biosystems). The abundance of target mRNAs was normalized by that of *Gapdh* mRNA. The PCR primers are listed in *Supplementary file 1C*.

### Protein extraction and immunoblot analysis

Total protein was extracted from mouse tissues by homogenization in a lysis solution consisting of 1 M Tris-HCL (pH 7.4), 0.1 M EDTA, 1% Nonidet P-40, 5 mM sodium pyrophosphate, and phosphatase (25955–24, Nacalai Tesque) and protease (07575–51, Nacalai Tesque) inhibitor cocktails. The homogenate was centrifuged at 15,000×*g* for 15 min at 4 °C, and the resulting supernatant was assayed for protein concentration with a Pierce BCA Protein Assay Kit (Thermo Fisher Scientific) and then subjected to SDS-polyacrylamide gel electrophoresis on a 10% acrylamide/4% bisacrylamide gel. The separated proteins were transferred to polyvinylidene difluoride membrane, which was then exposed to 5% dried skim milk for 1 hr at room temperature before incubation with primary antibodies, including those to PGC-1α (A12348, Abclonal), to UCP1 (sc-293418, Santa Cruz Biotechnology), and to GAPDH (CAB932Hu01, Cloud-Clone) for 16 hr at 4 °C. Immune complexes were detected with horseradish peroxidase–conjugated secondary antibodies (458, MBL; 7076P2, Cell Signaling Technology), enhanced chemiluminescence reagents (170–5061, Bio-Rad Laboratories, Inc), and an Amersham Imager 600 instrument (GE Healthcare). Band intensity was quantified with ImageJ software.

### Histology

BAT was fixed in formalin, sectioned, and stained with hematoxylin-eosin.

### Metabolomics analysis

Targeted profiling of amino acid metabolites was performed by capillary electrophoresis–time-of-flight mass spectrometry (CE-TOF/MS) with an Agilent 7100 CE system coupled to an 6230 TOF mass spectrometer (Agilent Technologies) according to previously described methods (*Okamatsu-Ogura et al., 2020*). The collected metabolite data were subjected to enrichment analysis using Metaboanalyst 6.0 (https://www.metaboanalyst.ca). Pathway enrichment was tested using the SMPDB databases.

### Measurement of noradrenaline

Tissue samples were homogenized in methanol containing internal standards (±)-Norepinephrine-D6 hydrochloride solution, Sigma-Aldrich. The mixture was centrifuged at 4 °C, and the aqueous fraction

was isolated and centrifuged through a cosmonice filter W (06543–04, Nacalai Tesque). Noradrenaline content of the supernatants was measured by high-performance liquid chromatography (SHIMAZU) with a reversed-phase column (Cadenza CD-C18 MF 100×2 mm 3 µm, Imtakt) and tandem mass spectrometry (MS/MS) (API 3200, AB SCIEX). Data are analyzed using AnalystLauncher (AB SCIEX).

## Measurement of [³H] leucine uptake

Mice were deprived of food for 3 hr before intravenous injection of L-[4,5-³H($N$)] leucine (NET1166, PerkinElmer) at a dose of 1 µCi/g. They were then maintained in a resting state for 40 min until blood and tissue collection, with perfusion being performed before tissue isolation. Isolated tissue was cut into small pieces, which were then placed in a round-bottom polypropylene tube before the addition of 500 µl of distilled water, 150 µl of 30% hydrogen peroxide (Wako), and 150 µl of 2 M KOH (1310-58-3, Hayashi Pure Chemical) in 2-propanol (000–64783, Kishida Chemical) and incubation at 65 °C for 2 hr. The tissue lysate was neutralized with 30% acetic acid, and 1 ml of liquid scintillation solubilizer (Soluen-350, PerkinElmer) was added to the sample before incubation at 50 °C for 5 hr and the further addition of 5 ml of liquid scintillation cocktail (Hionic-Flour, PerkinElmer). Leucine uptake in each organ was quantified by measurement of radioactivity with a scintillation counter (LSC-7400, Hitachi).

## Hepatic glycogen assay

The liver was homogenized in ice-cold citrate buffer (pH 4.2) containing NaF (2.5 g/l), the homogenate was centrifuged at 14,000×$g$ for 5 min at 4 °C, and the resulting supernatant was assayed for glycogen with an EnzyChrom Glycogen Assay Kit (BioAssay Systems).

## ELISAs

Blood samples were collected from the heart of mice that had been anesthetized by intraperitoneal injection of domitor (Zenoaq) at a dose of 0.75 mg/kg, midazolam (Wako) at 4 mg/kg, and butorphanol tartrate (Meiji Seika Pharma) at 5 mg/kg after food deprivation for 3 hr, and they were centrifuged at 7500×$g$ for 10 min at 4 °C to isolate serum. Serum IL-6 and TNF-α concentrations were determined with enzyme-linked immunosorbent assay (ELISA) kits (Legend Max Mouse IL-6 ELISA Kit, BioLegend, and Mouse TNF Alpha Uncoated ELISA, Invitrogen, respectively). Serum noradrenaline was extracted with the use of a cis-diol–specific affinity gel, acylated, enzymatically converted, and then measured with an ELISA kit (Noradrenaline Research ELISA, ImmuSmol). Serum corticosterone was determined with a Corticosterone ELISA Kit (Enzo Life Sciences).

## Statistical analysis

All experiments were independently replicated at least two times. Unless otherwise stated, all data were obtained from a minimum of n=3 biological replicates. Technical replicates are indicated in the figure legends.

Data are presented as means ± SEM unless indicated otherwise and were analyzed with statistical software (JMP or Statcel–The Useful Add-in Forms on Excel 4th edition). Comparisons between two groups were performed with the paired or unpaired t-test, as appropriate. Those among more than two groups were performed with Dunnett's test, by one-way analysis of variance (ANOVA) followed by the Tukey-Kramer test, or by two-way ANOVA followed by Tukey's post hoc test or the post hoc paired/unpaired t-test with Bonferroni's correction. A two-tailed p-value was calculated for all experiments, and a value of <0.05 was considered statistically significant.

## Acknowledgements

We thank members of the Radioisotope Center of Tokushima University Graduate School for technical support with the tracer experiments; members of the Special Mission Center for Metabolome Analysis, School of Medical Nutrition, Faculty of Medicine, Tokushima University, for technical support with the metabolomics analysis; Yoichiro Iwakura (University of Tokyo) for permission to study the IL-6 KO mice; Kohta Onishi (University of Tokushima) for technical support with the LC/MS/MS analysis; and the Support Center for Advanced Medical Sciences at Tokushima University Graduate School of Biomedical Sciences for general technical support. This study was supported by Funds for the Development of Human Resources in Science and Technology from the Ministry of Education, Culture, Sports, Science, and Technology of Japan (MEXT) through the Home for Innovative Researchers and

Academic Knowledge Users consortium, and by KAKENHI grants (21J14844 to YI-M, 23K19926 to YI-M, 22H03535 to HS, and 21K11724 to RT) from the Japan Society for the Promotion of Science.

## Additional information

### Funding

| Funder | Grant reference number | Author |
|---|---|---|
| Japan Society for the Promotion of Science | Grant-in-Aid for Scientific Research (KAKENHI) [Grant Number: 21J14844] | Yuna Izumi-Mishima |
| Japan Society for the Promotion of Science | Grant-in-Aid for Scientific Research (KAKENHI) [Grant Number: 22H03535] | Hiroshi Sakaue |
| Japan Society for the Promotion of Science | Grant-in-Aid for Scientific Research (KAKENHI) [Grant Number: 21K11724] | Rie Tsutsumi |
| Japan Society for the Promotion of Science | Grant-in-Aid for Scientific Research (KAKENHI) [Grant Number:23K19926] | Yuna Izumi-Mishima |

The funders had no role in study design, data collection and interpretation, or the decision to submit the work for publication.

### Author contributions

Yuna Izumi-Mishima, Conceptualization, Resources, Data curation, Formal analysis, Supervision, Funding acquisition, Validation, Investigation, Visualization, Methodology, Writing - original draft, Project administration, Writing – review and editing; Rie Tsutsumi, Conceptualization, Resources, Data curation, Formal analysis, Funding acquisition, Validation, Investigation, Visualization, Methodology, Writing - original draft, Writing – review and editing; Tetsuya Shiuchi, Conceptualization, Resources, Data curation, Formal analysis, Funding acquisition, Investigation, Visualization, Methodology, Writing - original draft, Writing – review and editing; Saori Fujimoto, Manaka Tsutsumi, Kazuhiro Nomura, Resources, Data curation, Formal analysis, Validation, Investigation, Methodology, Writing – review and editing; Momoka Taniguchi, Data curation, Formal analysis, Validation, Investigation, Methodology; Mizuki Sugiuchi, Data curation, Formal analysis, Validation, Investigation; Yuko Okamatsu-Ogura, Takeshi Yoneshiro, Masashi Kuroda, Resources, Methodology, Writing – review and editing; Hiroshi Sakaue, Conceptualization, Data curation, Formal analysis, Supervision, Funding acquisition, Validation, Investigation, Project administration, Writing – review and editing

### Author ORCIDs

Yuko Okamatsu-Ogura https://orcid.org/0000-0002-3008-9080
Hiroshi Sakaue https://orcid.org/0000-0002-2468-2363

### Ethics

All experiments were approved by the Animal Care and Use Committee of Tokushima University (protocol numbers: T2023-17 and T2020-63) and were conducted in accordance with the guidelines for the care and use of animals approved by the Council of the Physiological Society of Japan. All animal experiments were conducted in accordance with the ARRIVE guidelines.

Reviewer #1 (Public review): https://doi.org/10.7554/eLife.99982.3.sa1
Reviewer #2 (Public review): https://doi.org/10.7554/eLife.99982.3.sa2
Author response https://doi.org/10.7554/eLife.99982.3.sa3

## Additional files

### Supplementary files
MDAR checklist

Supplementary file 1. Primer sequences and additional data for the skeletal muscle immobilization model.

### Data availability
A dataset named 'Brown Adipose Tissue and Skeletal Muscle Coordinately Contribute to Thermogenesis in Mice' has been deposited in the Dryad Digital Repository (https://doi.org/10.5061/dryad.18931zd9c). It contains all the individual experiments used for the figures in the article.

The following dataset was generated:

| Author(s) | Year | Dataset title | Dataset URL | Database and Identifier |
|---|---|---|---|---|
| Izumi-Mishima Y, Tsutsumi R, Shiuchi T, Fujimoto S, Taniguchi M, Sugiuchi M, Tsutsumi M, Okamatsu-Ogura Y, Yoneshiro T, Kuroda M, Nomura K, Sakaue H | 2025 | Brown Adipose tissue and skeletal muscle Coordinately contribute to Thermogenesis in mice - Data set | https://doi.org/10.5061/dryad.18931zd9c | Dryad Digital Repository, 10.5061/dryad.18931zd9c |

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
