## [Editor Report · eLife Assessment]

This is a **useful** paper regarding the roles of brown adipose tissue and skeletal muscle in thermogenesis in mice, with potential significance for the field. The overall approach is innovative but on balance the evidence for the claim is **incomplete**, as cast immobilization, while innovative, is likely stressful, may impact muscle and BAT directly, and imposes an energetic cost of motion on the animal that is not accounted for. Further experiments are also needed to directly assess the role of adipose-derived BCAAs in thermogenesis. The authors have done a good job of textually editing their manuscript to clarify the findings and limitations of the study.

---

## [Referee Report · Reviewer #1 (Public review)]

Summary:

Heat production mechanisms are flexible, depending on a wide variety of genetic, dietary and environmental factors. The physiology associated with each mechanism is important to understand, since loss of flexibility associates with metabolic decline and disease.

The phenomenon of compensatory heat production has been described in some detail in publications and reviews, notably by modifying BAT-dependent thermogenesis (for example by deleting UCP1 or impairing lipolysis, cited in this paper).

These authors chose to eliminate exercise as an alternative means for maintaining body temperature. To do this, they cast either one or both mouse hindlimbs.

This paper is set up as an evaluation of a loss of function of muscle on the functionality of BAT. However, the authors show that cast immobilization (CI) does not work as a (passive) loss of function, instead this procedure produces a dramatic gain of function.

It does not test the hypothesis as stated, instead it adds an extraneous variable, which is that the animal is put under enormous stress, inducing b-adrenergic effectors, increased oxygen consumption, and IL6 expression in a variety of tissues, together with commensurate cachectic effects on muscle and fat. The BAT is stressed by this procedure, becoming super-induced but relatively poor functioning. This is an inaccurate experimental construct, and the paper is therefore full of wrong conclusions.

Within hours and days of CI, there is massive muscle loss (leading to high circulating BCAAs), and loss of lipid reserves in adipose and liver. The lipid cycle that maintains BAT thermogenesis is depleted and the mouse is unable to maintain body temperature.

I cannot agree with these statements in the Discussion -

"We have here shown that cast immobilization suppressed skeletal muscle thermogenesis, resulting in failure to maintain core body temperature in a cold environment."

• This result could also be attributed to high stress and decreased calorie reserves. Note also: CI suppresses 50% locomoter activity, but the actual work done by the mouse carrying bilateral casts is not taken into account (how heavy are they?). Presumably other muscles in the mouse body are compensating to allow the mouse to drag itself to the food source, to maintain food consumption, which remarkably, is unchanged. Is the demand for heat even the same when the mouse is wrapped in gypsum?

I cannot be convinced that this approach (CI) can be interpreted at all in terms of organ communication during thermogenic challenge. This paper describes instead the resilience and adaptation of mouse physiology in the face of dragging around hind limb casts.

From Rebuttal:

"On the other hand, the experiment shown in Fig.1C involved acute cold exposure of mice 2 h after cast immobilization. This result suggests that, even before the depletion of energy stores by immobilization of skeletal muscle, cast immobilization may cause cold intolerance in mice."

Since the mice are in acute recovery from the anesthetic, there can be no conclusions drawn about thermogenesis. Isoflurane is a great way to depress body temperature (http://www.ncbi.nlm.nih.gov/pubmed/12552204), and the recovery time is not known.

"In addition, as the reviewer suggests, cast immobilization may result in BAT thermogenesis and cachectic effects on muscle and fat. However, circulating corticosterone concentrations and hypothalamic CRH gene expression are not significantly altered after cast immobilization (Figure 2_figure supplement 2D-F)."

The absence of positive results from your stress assays does not exclude stress as the primary source of the results. These mice are not proceeding as normal with their lives - they are learning whole new behaviors in order to stay fed and watered.

---

## [Referee Report · Reviewer #2 (Public review)]

Summary:

In this study, the authors identified a previously unrecognized organ interaction where limb immobilization induces thermogenesis in BAT. They showed that limb immobilization by cast fixation enhances the expression of UCP1 as well as amino acid transporters in BAT, and amino acids are supplied from skeletal muscle to BAT during this process, likely contributing to increased thermogenesis in BAT. Furthermore, the experiments with IL-6 knockout mice and IL-6 administration to these mice suggest that this cytokine is likely involved in the supply of amino acids from skeletal muscle to BAT during limb immobilization.

Strengths:

The function of BAT plays a crucial role in the regulation of an individual's energy and body weight. Therefore, identifying new interventions that can control BAT function is not only scientifically significant but also holds substantial promise for medical applications. The authors have thoroughly and comprehensively examined the changes in skeletal muscle and BAT under these conditions, convincingly demonstrating the significance of this organ interaction.

Weaknesses:

Through considerable effort, the authors have demonstrated that limb-immobilized mice exhibit changes in thermogenesis and energy metabolism dynamics at their steady state. However, The impact of immobilization on the function of skeletal muscle and BAT during cold exposure has not been thoroughly analyzed.

Comments on revisions:

The authors appropriately responded to the reviewers' recommendations made during the previous round of peer review.

---

## [Author Response]

The following is the authors’ response to the current reviews.

**Response to eLife Assessment:**

We sincerely appreciate your recognition of the novelty and potential significance of our study, and we are grateful for your constructive and valuable comments.

With regard to your concern that cast immobilization (CI) may itself act as a stressor—potentially influencing skeletal muscle, brown adipose tissue (BAT), and locomotor energy expenditure—we fully recognize this as a highly important issue. In our study, we sought to interpret the findings in light of oxygen consumption and activity data; however, it is inherently difficult to disentangle systemic stress responses and the increased energetic costs associated with CI. We have therefore revised the manuscript to explicitly acknowledge this point as a limitation, and to identify it as a subject for future investigation.

We also greatly value your suggestion concerning the potential involvement of branched-chain amino acids (BCAAs) derived from adipose tissue in BAT thermogenesis. While our present work primarily focused on muscle-derived amino acids, previous studies have reported that impaired BCAA catabolism in white adipose tissue (WAT) is associated with elevated circulating BCAA levels and metabolic dysfunction [1]. Thus, the possibility that adipose tissue contributes to the BCAA pool used by BAT cannot be disregard. We fully agree that directly addressing this possibility would be highly valuable, and in future work we plan to locally administer isotope-labeled BCAAs into skeletal muscle or adipose tissue and analyze their contribution to circulating BCAA levels and BAT utilization. Although such experiments could not be performed within the timeframe of this resubmission, we have explicitly stated this limitation in the revised manuscript.

In summary, we have revised the text to acknowledge the limitations highlighted in your comments and to better clarify future research directions. We believe these revisions more accurately position our current study within the broader context. Once again, we are deeply grateful for your recognition of the originality of our work and for your constructive guidance in refining it.

**Response to Reviewers:**

We sincerely appreciate the reviewers’ thoughtful evaluations and constructive comments, and we are grateful for their recognition of the novelty and significance of our study.

**Response to Reviewer 1:**

We thank the reviewer for the detailed and thoughtful comments regarding the potential systemic effects of CI, including stress responses, energy balance, and tissue wasting. These factors are indeed critical when interpreting our findings, and we agree that CI is not merely a passive loss-of-function model but also introduces stress-related influences.

The principal aim of our study was to investigate the “physiological compensatory mechanisms” that are triggered by loss of muscle function induced by CI. Although CI inevitably elicits systemic metabolic alterations—including stress-related responses—our study is, to our knowledge, the first to demonstrate that a compensatory thermogenic pathway, mediated by the supply of amino acids from skeletal muscle to BAT, is activated under such conditions. We regard this as the central novelty of our work, and it is consistent with the reviewer’s observation that CI results in a “gain of function.”

Our intention is not to exclude stress as a contributing factor. Rather, we emphasize that under physiological stress conditions requiring BAT thermogenesis—such as reduced energy stores or decreased heat production from skeletal muscle—amino acid supply from muscle to BAT is induced. Importantly, this mechanism is not unique to CI, as we have confirmed similar metabolic crosstalk under acute cold exposure.

At the same time, we acknowledge that our current data do not allow us to conclude that “stress is not a primary driver” of BAT thermogenesis induced by CI. Chronic stress induced by CI appeared to be limited in our study (Fig. 2_figure supplement 2), but we cannot fully exclude stress-related effects. Accordingly, we now describe the potential triggers of BAT thermogenesis in the manuscript as either decreased body temperature due to muscle functional loss or stress, explicitly noting in the Discussion that stress and reductions in energy reserves may both contribute, as the reviewer suggested. We also modified the original overstatement that “suppression of muscle thermogenesis induces hypothermia,” and now limit the description to the observed phenomenon that “CI-induced restriction of muscle activity leads to reduced cold tolerance,” while recognizing that multiple factors—including stress, substrate availability, and BAT functional capacity—may underlie this effect.

We further appreciate the reviewer’s comment regarding the energetic burden imposed by CI. The cast weighed less than 2 g (5–10% of body weight), and thus increased locomotor costs cannot be excluded. However, locomotor activity during the dark phase was reduced by approximately 50%, making the net energetic effect difficult to quantify. In the manuscript, we now present oxygen consumption data and restrict our description to “an increase in oxygen consumption per body weight.” Moreover, as food intake remained almost unchanged compared with controls, the animals appear to have compensated for additional energetic demands, supporting the interpretation that the observed effects were not solely attributable to starvation.

We also find the reviewer’s suggestion—that CI induces BAT overactivation but impairs its functional capacity—extremely important. Indeed, although CI increased thermogenic gene expression in BAT, body temperature maintenance was impaired. We interpret this reduction in thermoregulation as reflecting decreased heat production from skeletal muscle; however, as the reviewer noted, under prolonged CI, depletion of energy stores could further prevent BAT from fully exerting its thermogenic function.

We have clarified in the revised Discussion that BAT activation under CI is transient, and that long-term outcomes may be influenced by contributions from other thermogenic organs, and that we recognize the impact of energy depletion as an important issue to be addressed in future studies. We also agree that detailed analyses of metabolic changes and BCAA dynamics following prolonged CI will be an important next step.

Regarding the reviewer’s concern about potential anesthesia effects on acute cold exposure experiments, we confirmed that body temperature had returned to baseline one hour before testing, and that mice displayed spontaneous feeding and grooming behaviors, which suggested adequate recovery. Moreover, the differences observed compared with sham-anesthetized controls support our interpretation that the results reflect CI-specific effects. Nonetheless, we acknowledge this potential confounding factor as an additional limitation.

**Response to Reviewer 2:**

We thank the reviewer for the constructive comments and clear summary of our findings. We fully agree that the impact of immobilization on skeletal muscle and BAT function under cold exposure represents a key future direction. In the present study, we performed acute cold exposure following short-term immobilization and assessed UCP1 expression and metabolic changes in BAT. However, we acknowledge that we did not fully examine coordinated functional adaptations between skeletal muscle and BAT under cold stress. In particular, how skeletal muscle–derived amino acid supply and IL-6–dependent mechanisms operate during cold exposure remains unresolved. We have therefore noted this explicitly as a limitation and highlighted it as a focus for future work. Going forward, we plan to investigate muscle–BAT metabolic crosstalk and IL-6 signaling in detail under cold conditions to clarify whether the observed responses are specific to CI or represent more general physiological adaptations.

(1) Herman MA, She P, Peroni OD, Lynch CJ, Kahn BB. Adipose tissue branched chain amino acid (BCAA) metabolism modulates circulating BCAA levels. J Biol Chem. 2010;285(15):11348-56. doi:10.1074/jbc.M109.075184.

The following is the authors’ response to the original reviews.

**Reviewer #1 (Public Review):**
Summary:Heat production mechanisms are flexible, depending on a wide variety of genetic, dietary, and environmental factors. The physiology associated with each mechanism is important to understand since loss of flexibility is associated with metabolic decline and disease. The phenomenon of compensatory heat production has been described in some detail in publications and reviews, notably by modifying BAT-dependent thermogenesis (for example by deleting UCP1 or impairing lipolysis, cited in this paper). These authors chose to eliminate exercise as an alternative means of maintaining body temperature. To do this, they cast either one or both mouse hindlimbs. This paper is set up as an evaluation of a loss of function of muscle on the functionality of BAT.Strengths:The study is supported by a variety of modern techniques and procedures.Weaknesses:The authors show that cast immobilization (CI) does not work as a (passive) loss of function, instead, this procedure produces a dramatic gain of function, putting the animal under considerable stress, inducing b-adrenergic effectors, increased oxygen consumption, and IL6 expression in a variety of tissues, together with commensurate cachectic effects on muscle and fat. The BAT is put under considerable stress, super-induced but relatively poor functioning. Thus within hours and days of CI, there is massive muscle loss (leading to high circulating BCAAs), and loss of lipid reserves in adipose and liver. The lipid cycle that maintains BAT thermogenesis is depleted and the mouse is unable to maintain body temperature.I cannot agree with these statements in the Discussion:"We have here shown that cast immobilization suppressed skeletal muscle thermogenesis, resulting in failure to maintain core body temperature in a cold environment."This result could also be attributed to high stress and decreased calorie reserves. Note also: CI suppresses 50% of locomotor activity, but the actual work done by the mouse carrying bilateral casts is not taken into account.

We appreciate the reviewer's suggestion. We thank you for raising this issue. As the reviewers suggest, we also consider that cold intolerance resulting from cast immobilization may be attributed to high stress levels, decreased calorie reserves, or reduced systemic locomotor activity. Indeed, reductions in the weight of visceral adipose tissue weight and increases in lipid utilization were observed in the early phase of cast immobilization (Fig.2G and 2F). This suggests that the depletion of calorie reserves induced by stress may affect cold intolerance in cast immobilized mice (Fig.1A-1B). On the other hand, the experiment shown in Fig.1C involved acute cold exposure of mice 2 h after cast immobilization. This result suggests that, even before the depletion of energy stores by immobilization of skeletal muscle, cast immobilization may cause cold intolerance in mice. In addition, as the reviewer suggests, cast immobilization may result in BAT thermogenesis and cachectic effects on muscle and fat. However, circulating corticosterone concentrations and hypothalamic CRH gene expression are not significantly altered after cast immobilization (Figure 2_figure supplement 2D-F). This raises questions about the contribution of stress to the changes in the systemic energy metabolism in this model. As such, we responded to the reviewers’ comments by revising this statement at the beginning of the ‘Discussion’ section and adding a discussion on pages 16, in addition to the existing discussion on pages 17–18.

Furthermore, to respond as best we could to the reviewer's comments, we performed additional experiments using the restraint stress model (Figure 7). We found that short-term restraint stress may recruit substrate supply from skeletal muscle for BAT thermogenesis via Il6 gene expression. Based on these data, we speculate that the interaction between BAT and skeletal muscle amino acid metabolism may operate under various physiological stress conditions, including infection and exercise, as well as skeletal muscle immobilization, stress, and cold exposure. This interaction may play a significant role in regulating body temperature and energy metabolism. We are currently investigating the effects of sympathetic activation on skeletal muscle amino acid metabolism and systemic thermoregulation via IL-6 secretion from skeletal muscle using a new model. These data will be reported in a subsequent report.

"Thermoregulatory system in endotherms cannot be explained by thermogenesis based on muscle contraction alone, with nonshivering thermogenesis being required as a component of the ability to tolerate cold temperatures in the long term."This statement is correct, and it clearly showcases how difficult it is to interpret results using this CI strategy. The question to the author is- which components of muscle thermogenesis are actually inhibited by CI, and what is the relative heat contribution?

We appreciate raising this important issue. This study required the measurements of skeletal muscle temperature and electromyography in mice with cast immobilization, but we were unable to perform these measurements. We have therefore described the reviewers suggest on page 18 as limitations of this study.

In our additional experiments, we found that several genes that are usually activated in skeletal muscle during cold exposure are repressed in mice with cast immobilization (Figure 1_figure supplement 1_G-1K). Skeletal muscle is an important thermogenic organ. Although the role of the sarcolipin gene in non-shivering thermogenesis is well understood, the primary regulator of thermogenesis in metabolism and shivering remains unclear. In Future, we would like to use models in which key signals for energy metabolism are inhibited, such as muscle-specific PGC-1α-deficient mice and muscle-specific AMPK-deficient mice, to clarify important factors in skeletal muscle heat thermogenesis. We expect this approach to enable us to analyze the relative thermal contributions of each component of the heat production process in skeletal muscle, which has proven difficult in immobilized muscle models.

This conclusion is overinterpreted:"In conclusion, we have shown that cast immobilization induced thermogenesis in BAT that was dependent on the utilization of free amino acids derived from skeletal muscle, and that muscle-derived IL-6 stimulated BCAA metabolism in skeletal muscle. Our findings may provide new insights into the significance of skeletal muscle as a large reservoir of amino acids in the regulation of body temperature".In terms of the production of the article - the data shown in the heat maps has oddly obscure log10 dimensions. The changes are minimal, approx. 1.5x increase/decrease and therefore significance would be key to reporting these data. Fig.3C heatmap is not suitable. What are the 6 lanes to each condition? Overall, this has little/no information.Rather than cherry-picking for a few genes, the results could be made more rigorous using RNA-seq profiling of BAT and muscle tissues.

We agree that this is an important point. Indeed, our model of skeletal muscle immobilization reveals only modest changes in metabolomics and gene expression analysis. We consider this to be a weakness of the study. However, the interactive thermogenic system that we discovered between skeletal muscle and BAT may also function under other conditions, such as acute stress and cold exposure. We should investigate this further in future models involving such dramatic metabolic changes. In fact, it has been shown that the levels of several metabolites are significantly altered in BAT after acute cold exposure.[1] Therefore, we have corrected the conclusion of this section, as stated on page 18, and added it. We also performed an enrichment analysis on the metabolomics data in BAT following cast immobilization and included the results in Figure 2_figure Supplement 1A. In addition, we excluded the heatmap from Fig. 3C of the pre-revision manuscript, as advised by the reviewer. Although we excluded the results in Figure 3C, we consider Figure 3_figure supplement_1 to be sufficient for the text.

In addition, we agree with the reviewer's remarks on our gene expression analysis. In this study, we were unable to examine RNA-seq profiling of BAT and muscle tissue. Therefore, we have described this as a limitation of the study on page 20. However, we are interested in investigating the effect of IL-6 derived from skeletal muscle on RNA-seq profiling of skeletal muscle and BAT. We will conduct future RNA-seq analyses of BAT and skeletal muscle, using models of skeletal muscle immobilization, acute cold exposure and restraint stress.

**Reviewer #2 (Public Review):**
Summary:In this study, the authors identified a previously unrecognized organ interaction where limb immobilization induces thermogenesis in BAT. They showed that limb immobilization by cast fixation enhances the expression of UCP1 as well as amino acid transporters in BAT, and amino acids are supplied from skeletal muscle to BAT during this process, likely contributing to increased thermogenesis in BAT. Furthermore, the experiments with IL-6 knockout mice and IL-6 administration to these mice suggest that this cytokine is likely involved in the supply of amino acids from skeletal muscle to BAT during limb immobilization.Strengths:The function of BAT plays a crucial role in the regulation of an individual's energy and body weight. Therefore, identifying new interventions that can control BAT function is not only scientifically significant but also holds substantial promise for medical applications. The authors have thoroughly and comprehensively examined the changes in skeletal muscle and BAT under these conditions, convincingly demonstrating the significance of this organ interaction.Weaknesses:Through considerable effort, the authors have demonstrated that limb-immobilized mice exhibit changes in thermogenesis and energy metabolism dynamics at their steady state. However, The impact of immobilization on the function of skeletal muscle and BAT during cold exposure has not been thoroughly analyzed.
**Reviewer #3 (Public Review):**
Summary:In this manuscript, the authors show that impairment of hind limb muscle contraction by cast immobilization suppresses skeletal muscle thermogenesis and activates thermogenesis in brown fat. They also propose that free BCAAs derived from skeletal muscle are used for BAT thermogenesis, and identify IL-6 as a potential regulator.Strengths:The data support the conclusions for the most part.Weaknesses: The data provided in this manuscript are largely descriptive. It is therefore difficult to assess the potential significance of the work. Moreover, many of the described effects are modest in magnitude, questioning the overall functional relevance of this pathway. There are no experiments that directly test whether BCAAs derived from adipose tissue are used for thermogenesis, which would require more robust tracing experiments. In addition, the rigor of the work should be improved. It is also recommended to put the current work in the context of the literature.

We appreciate the reviewer's valuable feedback. As the reviewer pointed out, many of the effects described in this study are modest in magnitude. This reflects a limitation of our study, which used skeletal muscle immobilization as a model. To clarify the overall functional relevance of this pathway, we therefore plan to use alternative models in which BAT thermogenesis and systemic cachectic effect are more strongly induced. We have added this point to the 'Conclusion' section on page 18.

In addition, previous findings reported that mitochondrial BCAA catabolism in brown adipocytes promotes systemic BCAA clearance, suggesting that BCAAs may be supplied to BAT from other organs during BAT thermogenesis.[5] However, as the reviewer rightly pointed out, the current study did not directly investigate whether BCAAs derived from adipose tissue contribute to thermogenic processes. In light of this, we have revised the manuscript to include a statement in the limitations section on page 20 that addresses this point.

Metabolomic analysis of white adipose tissue (WAT) following skeletal muscle immobilization revealed alterations in amino acid concentrations in WAT in response to cast immobilization (Author response image 1A). Notably, levels of BCAAs in WAT remained largely unchanged at 24 hours after cast immobilization, but increased significantly by day 7 (Author response image 1B). At the 24-hour time point, when BAT thermogenesis is known to be activated, WAT weights was found to be reduced (Fig. 2H). Gene expression analysis of amino acid metabolism-related genes in WAT at this time point revealed a modest upregulation of several genes (Author response image 1C). Furthermore, a slight increase in the uptake of [^3^H] leucine into WAT was observed following immobilization (Fig. 3C). Collectively, these findings suggest that BCAAs within WAT may be primarily metabolized locally rather than being mobilized and supplied to BAT. In addition, given the relatively low levels of BCAAs per tissue mass and the limited capacity for BCAA uptake in WAT compared to other tissues, we consider it unlikely that WAT serves as a major reservoir of BCAAs.

(A) Amino acids in epididymal white adipose tissue (eWAT) of IL-6 KO (–/–) and WT (+/+) mice without (control) or with bilateral cast immobilization for the indicated times. Results are presented as heat maps of the log10 value of the fold change relative to control WT mice and are means of four mice in each group. (B) BCAA concentrations in eWAT of IL-6 KO and WT mice without (control) or with bilateral cast immobilization for 1 or 7 days. (n = 4 per group) (C) RT and real-time PCR analysis of the expression of SLC1A5, SLC7A1, SLC38A2, SLC43A1, BCAT2 and BCKDHA genes in eWAT of mice without (control) or with bilateral cast immobilization for 24 h. (n = 6 per group). All data other than in (A) are means ± SEM. *p < 0.05, **p < 0.01, ***p < 0.001 as determined by Dunnett's test (B) or by the unpaired t test (C).

**Reviewer #1 (Recommendations for the authors):**
• Gypsum is an irrelevant label. Label consistently, with a procedure acronym, like CI or Imm.

We apologize for any confusion that our notation may have caused. We corrected all labels relating to the skeletal muscle immobilization model in mice to 'Imm'.

There are many grammatical errors and typos. Search for an example on Fudure1. The sense of some sentences is enough to obscure their meaning.

We appreciate the reviewer's points. We have checked the article for grammatical and typographical errors, correcting them where necessary.

• Figures 6E and F need to be re-annotated in the legend and on figures.

Following the peer reviewer's advice, we have re-annotated the Figure legends of this result.

**Reviewer #2 (Recommendations for the authors):**
(1) It is difficult to understand how the data presented in Supplemental Table 1 were obtained. This appears to be data showing that the skeletal muscle weight of the hind limbs in mice accounts for 40 to 50% of the total skeletal muscle weight. How did the authors calculate the muscle weight? Specifically, how did they measure the weight of muscles that are neither in the hind limbs nor in the forelimbs ("Other")? Was this estimated from whole-body CT or MRI data?In the legend, it mentions "the posterior cervical region," but what exactly was measured in the posterior cervical region? The methods for this data should be clearly described.

We appreciate the reviewers' comments. We apologize for any confusion caused by inadequate explanation of this data. This data was obtained by removing skeletal muscle from the posterior cervical region and measuring the weight of the wet tissue. We have taken care to remove most of the skeletal muscle, but some will remain. However, we do not believe that these errors are significant enough to alter the interpretation of the results. This has now been added to the 'Methods' section on page 21.

(2) Through considerable effort, the authors have demonstrated that limb-immobilized mice exhibit changes in thermogenesis and energy metabolism dynamics at their steady state. However, it remains unclear why limb-immobilized mice have reduced tolerance to cold exposure. Was there any change in the abundance of energy metabolism-related genes during cold exposure between the immobilized and control mice? For example, if the gene expression of UCP1 and UCP2, which are typically upregulated in brown adipose tissue (BAT) and skeletal muscle during cold exposure, was suppressed in the immobilized mice, it might explain their reduced cold tolerance. Thus, the changes in the response of skeletal muscle and BAT to cold exposure between immobilized and control mice should also be analyzed.

We thank the reviewer for the constructive comments. We consider the main weakness of this study to be the fact that we were unable to measure the temperature and electromyography (EMG) of the skeletal muscles of the cast-immobilized mice. Following the reviewers' advice, we analyzed the expression levels of several genes related to heat production or energy metabolism (Ucp1, Ucp2, Ucp3, Sln and Ppargc1a) in BAT and skeletal muscle of cast-immobilized mice after acute cold exposure (Figure1_figure supplement 1G-1K). The results showed that the expression of several genes that are usually increased in BAT and skeletal muscle during cold exposure was repressed in cast-immobilized mice. Notably, cast immobilization did not induce the UCP2 and PGC-1α genes at room temperature, and their upregulation during cold exposure was also suppressed in cast-immobilized mice. UCP2 is known to alter its expression in relation to energy metabolism, but it is unclear whether it regulates energy metabolism.[2] Additionally, UCP2 is understood to play a non-role in thermogenesis, and the function of the UCP2 in skeletal muscle remains unclear.[3] On the other hands, PGC-1α is widely recognized as a transcriptional coactivator that regulates various metabolic processes, including thermogenesis.[4] In our study, we found that the amounts of metabolites in the TCA cycle and the expression of the PGC-1α gene were decreased rapidly in immobilized skeletal muscle. This suggests that the metabolic rate is reduced in immobilized skeletal muscle (Figure 1_figure supplement 2A and 2F). In endothermic animals, energy expenditure in skeletal muscle plays a significant role in maintaining body temperature during both activity and rest. Hence, it is assumed that the reduced metabolic rate in skeletal muscle significantly impacts the maintenance of body temperature in cold conditions. Further investigation is required into the function of these genes in skeletal muscle thermogenesis, but we expect that the additional data suggest that the loss of muscle function due to immobilization affects the maintenance of body temperature under cold temperature. These results were discussed further on page 15.

**Reviewer #3 (Recommendations for the authors):**
There are also more specific concerns related to the data supporting the claims.(1) The relevance of increasing thermogenesis in BAT after cast immobilization is unclear, as adult humans have very little BAT. Thermogenesis gene and protein expression should be measured in white adipose tissue.

We would like to thank the reviewers for highlighting this important issue. We agree with the reviewer's comments. We did not observe significant changes in UCP1 expression in the subcutaneous adipose tissue of the inguinal region following skeletal muscle immobilization. We suspect that this is because skeletal muscle immobilization in mice did not exert a strong enough effect to induce browning of white adipose tissue. The ability of immobilizing skeletal muscle to activate thermogenesis in brown or beige adipocytes in adults remains unclear. We have therefore noted this limitation in our study in line 6.

Additionally, in this study, we aimed to clarify the role of skeletal muscle as an amino acid reservoir under metabolic stress conditions that increase BAT thermogenesis. To this end, we employed models of skeletal muscle immobilization, acute cold exposure, and restraint stress. We also intend to analyze the metabolic interactions between beige adipose tissue and skeletal muscle in more detail using models that induce browning, such as exercise or cold acclimation.

(2) In Figures 1E-G, there is no significant difference in UCP1 levels relative to the control, but body temperature is lowered from day 2 to day 7. How do the authors explain this?

This is an important point. We consider the decrease in body temperature of mice following cast immobilization at room temperature to be the result of a reduction in systemic locomotor activity.

(3) The small induction of PGC1a seen at 10 hours goes away after day 3. Why is this?

This is an important point. Our investigation showed that the norepinephrine concentration in BAT and blood of cast-immobilized mice tends to increase, peaking at 24 hours of immobilization (Fig. 1H and Figure 2_figure supplement 2D), and then gradually returns to baseline. We speculate that this transient activation of the sympathetic nervous system may affect the expression of PGC1α in BAT. Additionally, although thermogenesis in BAT temporarily increases after skeletal muscle immobilization, studies from other research groups suggest that long-term skeletal muscle immobilization (two weeks) may increase non-shivering thermogenesis in skeletal muscle via high expression SLN.[6] Therefore, we hypothesize that other thermogenic mechanisms besides BAT might be involved during prolonged cast immobilization. We have added a discussion of these topics on page 16.

(4) The metabolic cage data are marked in multiple places as significant, but the effect size is extremely small. Please describe how significance was calculated (Figure 5 supplement 1B, E, F).

This is a valid point. This data was statistically analyzed using daily averages, with the results then being compiled. However, the figure was amended because it was not appropriate to use the original to demonstrate significant differences.

(5) How does IL-6 increase BCAA levels in muscle?

This is an important point. We are also investigating this issue with great interest. In future, we will use RNA-seq profiling to investigate the mechanism by which IL-6 regulates amino acid metabolism in skeletal muscle. This point was added as a

limitation of the study on page 19.

(6) What is the mechanism behind the elevated il6 levels after cast immobilization?

We appreciate the reviewer's points. Since IL-6 gene expression in skeletal muscle increases in response to acute cold exposure and acute stress, we hypothesize that IL-6 is regulated by β-adrenergic effectors. In our preliminary experiments, stimulation with norepinephrine or with clenbuterol, a β2-adrenergic receptor agonist, suggests an increase in IL-6 gene expression and the intracellular free BCAA concentration in cultured mouse muscle cells (Author response image 2A-2D). Going forward, our plans include conducting further studies using a mouse model in which the sympathetic nervous system is activated by administering LPS intracerebroventricularly, as well as using muscle-specific β2-adrenergic receptor knockout mice.

Reference:

(1) Okamatsu-Ogura, Y., et al. UCP1-dependent and UCP1-independent metabolic changes induced by acute cold exposure in brown adipose tissue of mice. Metabolism. 2020 113: 154396 doi: 10.1016/j.metabol.2020.154396.

(2) Patrick Schrauwen and Matthijs Hesselink, UCP2 and UCP3 in muscle controlling body metabolism., J Exp Biol. 2002 Aug;205(Pt 15):2275-85. doi: 10.1242/jeb.205.15.2275.

(3) C Y Zhang, et al., Uncoupling protein-2 negatively regulates insulin secretion and is a major link between obesity, beta cell dysfunction, and type 2 diabetes., Cell. 2001 Jun 15;105(6):745-55. doi: 10.1016/s0092-8674(01)00378-6.

(4) Christophe Handschin and Bruce M Spiegelman, Peroxisome proliferator-activated receptor gamma coactivator 1 coactivators, energy homeostasis, and metabolism., Endocr Rev. 2006 Dec;27(7):728-35. doi: 10.1210/er.2006-0037.

(5) Yoneshiro, et al., BCAA catabolism in brown fat controls energy homeostasis through SLC25A44. Nature. 2019 572(7771): 614-619 doi: 10.1038/s41586-019-1503-x.

(6) Shigeto Tomiya, et al., Cast immobilization of hindlimb upregulates sarcolipin expression in atrophied skeletal muscles and increases thermogenesis in C57BL/6J mice., Am J Physiol Regul Integr Comp Physiol. 2019 Nov1;317(5):R649-R661.doi:10.1152/ajpregu.00118.2019.